# Choice selective inhibition drives stability and competition in decision circuits

James P. Roach [1,2], Anne K. Churchland [2] & Tatiana A. Engel [1]✉

During perceptual decision-making, the firing rates of cortical neurons reflect upcoming choices. Recent work showed that excitatory and inhibitory neurons are equally selective for choice. However, the functional consequences of inhibitory choice selectivity in decision-making circuits are unknown. We developed a circuit model of decision-making which accounts for the specificity of inputs to and outputs from inhibitory neurons. We found that selective inhibition expands the space of circuits supporting decision-making, allowing for weaker or stronger recurrent excitation when connected in a competitive or feedback motif. The specificity of inhibitory outputs sets the trade-off between speed and accuracy of decisions by either stabilizing or destabilizing the saddle-point dynamics underlying decisions in the circuit. Recurrent neural networks trained to make decisions display the same dependence on inhibitory specificity and the strength of recurrent excitation. Our results reveal two concurrent roles for selective inhibition in decision-making circuits: stabilizing strongly connected excitatory populations and maximizing competition between oppositely selective populations.

Perceptual decision-making requires neural circuits to integrate evidence and classify a stimulus to trigger the correct behavioral response. Neurons in a range of cortical areas modulate their firing rate to signal animal's choice[1]. The functional properties of decision-making neural circuits have been extensively studied and modeled[2–9]. Central to the function of these circuit models are attractors in the activity space which characterize the population's encoding of a given choice. The attractor mechanism driving the decision-making activity in these models relies on structured recurrent connections between populations of excitatory neurons that are each selective for a different choice[8,10,11]. Inhibitory neurons, in this view, are merely supporting actors facilitating competition and providing balance to the excitatory neurons.

Since the canonical models of decision-making circuits were built, the diversity and complexity of inhibitory neurons within the cortex have been characterized in increasing detail[12]. In primary sensory areas, inhibitory neurons are generally more broadly tuned[13] and more densely connected to neighboring excitatory neurons[14,15]. These inhibitory neurons reliably modulate spike output to reflect

stimulus features and have highly specific connectivity to surrounding excitatory neurons[16,17]. The stimulus selectivity of inhibitory neurons is enhanced by learning and attention[18] suggesting that task dependent modulation of inhibitory activity is necessary for cognition. Beyond the primary sensory cortex, stimulus information and animal choice can be decoded from the activity of inhibitory neurons in secondary sensory and association areas indicating a role for selective inhibition in higher cognitive functions, such as decision making[19–21]. While there is growing evidence that the activity and connectivity of inhibitory neurons is as complex as excitatory neurons, how the selectivity of inhibitory activity and the diversity of their connections affect the decision-making function of cortical circuits is still unknown.

To reveal the role of choice selective inhibitory neurons in decision-making computations we extended a well established mean-field model of decision-making circuits[4] to account for the presence of inhibitory choice selectivity. Our model allows us to parametrically alter the specificity of connections between two choice-selective excitatory and two choice-selective inhibitory populations. Through

[1]Cold Spring Harbor Laboratory, Cold Spring Harbor, NY, USA. [2]Department of Neurobiology, David Geffen School of Medicine, University of California Los Angeles, Los Angeles, CA, USA. ✉e-mail: engel@cshl.edu

analysis of this model, we found that while inhibition must drive competition between choice-selective excitatory populations it must also stabilize activity driven by recurrent excitation at the same time. These two concurrent roles are mediated by inhibitory connections to the excitatory populations and either role can be enhanced by structured inhibitory connectivity. We found that inhibitory selectivity expands the space of possible circuits which support decision-making by enhancing either a competitive or stabilizing role for inhibition. In addition, the connectivity motif between choice selective populations alters the underlying attractor dynamics and modulates the decision-making performance to prioritize speed or accuracy. We generalized these results by training recurrent neural networks (RNNs) to perform the same decision-making task. After training, RNNs had both excitatory and inhibitory units significantly selective for choice and displayed a similar dependence between the specificity of excitatory and inhibitory connections found in the mean-field model. Finally, we perturbed inhibitory neuron activity in these models to probe the dynamical regime in which the circuit operates. We found two regimes in which circuits respond differently to perturbations of inhibitory neurons: one in which the competitive role dominates and the other in which the stabilizing role dominates. Our work demonstrates that choice selective inhibition impacts decision-making behavior by enhancing either the competitive or the stabilizing role for inhibition in the circuit. These results generate testable predictions for perturbation experiments.

## Results

We consider circuits where two excitatory (E) populations integrate dedicated streams of sensory evidence to produce a categorical choice (Fig. 1a). In contrast to previous circuit models of decision-making with global inhibition, we include two inhibitory (I) populations which can inherit choice selectivity from excitatory neurons (Methods). We model the circuit dynamics using two-dimensional mean-field equations where the mean presynaptic activation of N-methyl-D-aspartate (NMDA) receptor of the two excitatory ($E_1$ and $E_2$) populations are the dynamic variables[4]. The average strength of connections between the four choice selective populations is controlled by a specificity parameter $\gamma$. For each of three connection classes (E to E, E to I, and I to E; Fig. 1b), $\gamma_{EE}$, $\gamma_{EI}$, and $\gamma_{IE}$ set the balance of connection strengths between populations with the same and opposite choice selectivity (Fig. 1c). For example, $(1 + \gamma_{EE})$ is the strength of feedback connections within excitatory populations selective for the same choice, and $(1 - \gamma_{EE})$ is the strength of connections between excitatory populations selective for the opposite choice. We keep $\gamma_{EE}$ positive due to the importance of recurrent feedback excitation in the function of these circuits[4]. When $\gamma_{EE} = 1$, each of $E_1$ and $E_2$ have a strong self-excitatory feedback and are not connected to each other. When $\gamma_{EE} = 0$, the

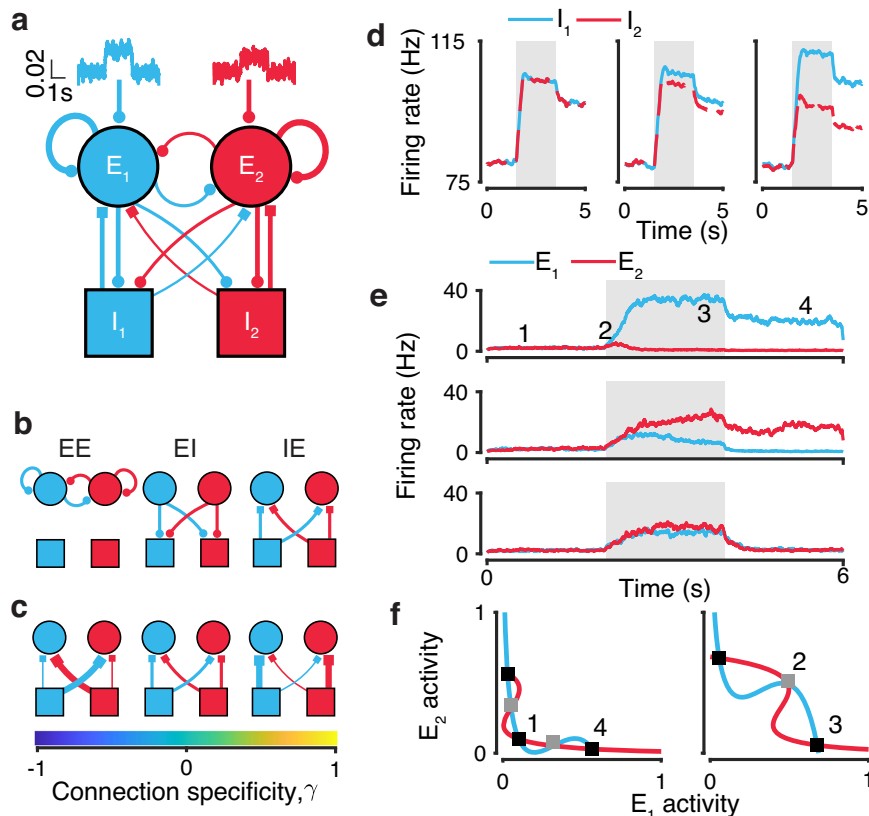

**Fig. 1 | A mean-field circuit model of decision making with choice-selective inhibition. a** The circuit diagram of the model with choice-selective excitatory and inhibitory populations. **b** The circuit model includes three connection classes: excitatory-excitatory (EE), excitatory-inhibitory (EI), and inhibitory-excitatory (IE). **c** The parameter $\gamma$ controls the specificity of connections between choice-selective populations. The output connections preferentially target neurons with the same choice preference when $\gamma$ is positive, and with the opposite choice preference when $\gamma$ is negative. **d** $\gamma_{EI}$ controls inhibitory choice selectivity. Firing rate of inhibitory populations for $\gamma_{EI} = 0$ (left), $\gamma_{EI} = 0.05$ (center), $\gamma_{EI} = 0.25$ (right) are shown for an example trial with stimulus strength equal to 20. **e** Circuits report choices by elevating the firing rate of one excitatory population. Example trials showing $E_1$ (blue) and $E_2$ (red) population activity for stimulus strength equal to 20 (upper panel), and for stimulus strength equal to 0 on a completed (middle panel) and invalid trial (lower panel). Gray shading indicates stimulation period. Numbers indicate activity corresponding to fixed points in **f**. **f** Eight fixed points are required for decision-making dynamics in the circuit: five in the unstimulated phase-plane (left) and three in the stimulated phase-plane (right, stimulus strength is equal to 0). Lines show nullclines of $E_1$ and $E_2$ populations, black squares indicate fixed-point attractors, and gray squares indicate saddle-points for a circuit with nonselective inhibition.

strengths of excitatory connections between and within $E_1$ and $E_2$ are all equal. Inhibitory choice selectivity is controlled by $\gamma_{EI}$ defined in the same way, which is also positive because inhibitory neurons inherit choice and stimulus information from the excitatory neurons. When $\gamma_{EI} = 1$, inhibitory population $I_1$ receives excitatory inputs from $E_1$ but not $E_2$ and vice versa. When $\gamma_{EI} = 0$, each $I_1$ and $I_2$ receive equal excitatory inputs from $E_1$ and $E_2$. Thus, inhibitory activity is not choice selective when $\gamma_{EI} = 0$ because inhibitory neurons receive equal input from both excitatory populations. Inhibitory choice selectivity emerges as $\gamma_{EI}$ increases (Fig. 1d).

For inhibitory choice selectivity to have any effect on circuit function, the outputs of inhibitory populations must be structured (i.e. $\gamma_{IE} \neq 0$; Fig. 1c). The specificity of inhibitory outputs $\gamma_{IE}$ can range between $[-1, 1]$ with negative values favoring connections between E and I populations with opposite choice preference and positive values favoring connections between E and I populations with the same choice preference. When $\gamma_{IE} = 1$, $I_1$ sends inhibitory output to $E_1$ but not $E_2$. When $\gamma_{IE} = -1$, $I_1$ sends inhibitory output to $E_2$ but not $E_1$. Thus, the specificity of inhibitory output connectivity defines three circuit motifs: contraspecific for $\gamma_{IE} < 0$, ipsispecific for $\gamma_{IE} > 0$, and nonspecific for $\gamma_{IE} = 0$.

In any decision-making circuit, inhibition concurrently fulfills two roles. The first is providing the substrate for competition between the excitatory populations, and the second is stabilizing the self-amplification driven by strongly recurrent excitatory populations. Both of these roles must be fulfilled for a circuit to function, but specific connections to and from inhibitory populations could enhance one of these roles (Fig. 1c). Specifically, ipsispecific inhibition can promote stabilizing feedback and contraspecific inhibition can maximize competition.

In response to an input stimulus, the circuit can produce different choice outcomes by changing the firing rates of the excitatory populations. Circuits report a choice by persistently raising the firing rate of one excitatory population at least 15 Hz above the other. Trials where this separation does not occur are considered invalid and not included in the calculation of psychometric or chronometric functions (Fig. 1e, Methods). We also require that prior to the stimulus onset, the circuit maintains low, symmetric activation of excitatory neurons. Persistence of the decision after stimulus offset allows for a choice readout to be made even after a significant delay and its utility led us to include the working memory of a choice in our criteria for inclusion as a circuit supporting decision-making (Fig. 1e). These dynamics are governed by eight fixed points across the phase planes of unstimulated and stimulated system, which are essential for the functional decision-making and working memory behavior (Fig. 1e, f). Prior to stimulus onset, both excitatory populations maintain low symmetric activation, which is set by an attractor located near the origin in the unstimulated phase plane. Following stimulus onset, the firing rate for both populations increases as the system approaches a saddle point along the stable manifold which acts as a separatrix between two choice attractors in the stimulated phase plane. Following stimulus offset, the system returns to its unstimulated phase plane and the choice of the circuit is preserved by one of two working memory attractors.

## Inhibitory connection specificity expands the space of circuits that support decision making

Using the mean-field model, we investigated how the circuit's ability to perform decision-making depends on the inhibitory connectivity structure. Specifically, we determined how choice-selective inhibition affects the presence of the eight fixed points (three attractors and two saddle points in the unstimulated phase plane, and two attractors and one saddle in the stimulated phase plane) governing decision-making behavior. We sampled the specificity parameter space to identify circuits which support these eight fixed points (Fig. 2a). We found that a

broad range of circuit configurations can support decision making. There are two components of inhibitory choice selectivity which rely on specific connections to and from inhibitory populations. The first is the degree of choice selective firing by inhibitory neurons that is controlled by $\gamma_{EI}$. The second is the degree to which inhibitory populations have a specific effect on excitatory neurons that is controlled by $\gamma_{IE}$. We combine these two components into a specificity index $\gamma_{EI}\gamma_{IE}$, which is negative for contraspecific and positive for ipsispecific circuits following the sign of $\gamma_{IE}$. The specificity of excitatory and inhibitory connections is highly correlated in circuits supporting decision making (Fig. 2b). When inhibition is nonselective ($\gamma_{EI} = 0$) or nonspecific ($\gamma_{IE} = 0$), the strength of recurrent excitation ($\gamma_{EE}$) is highly constrained and deviations from a narrow range leads to the loss of one of the essential fixed points (Fig. 2c). For circuits with selective inhibition, a wider range of $\gamma_{EE}$ will support decision making as long as a complementary inhibitory motif is present. For low $\gamma_{EE}$, the inhibitory motif must be contraspecific ($\gamma_{EI}\gamma_{IE} < 0$, Fig. 2b and d left) and for high $\gamma_{EE}$ it must be ipsispecific ($\gamma_{EI}\gamma_{IE} > 0$, Fig. 2b, d right). A contraspecific inhibitory motif can promote competition in circuits where excitatory feedback connections are insufficiently strong to amplify firing rate differences between choice selective populations. An ipsispecific inhibitory motif can stabilize excitatory feedback to prevent inadvertent winner-take-all dynamics in the absence of stimulus in circuits with strong excitatory specificity. By enhancing either the competitive or stabilizing role, circuits with choice selective inhibitory populations can support decision making for a wider range of $\gamma_{EE}$ (Fig. 2b).

The emphasis on competition or stability can also be seen in which fixed points are lost when connection specificity between excitatory and inhibitory populations are not complementary. When $\gamma_{EE}$ is low, nonspecific and ipsispecific circuits lack the fixed points representing choice both in the presence and absence of stimulation as well as the saddle point during the stimulus (Fig. 2d left, Supplementary Fig. 1), because recurrent excitation is too weak to drive competition alone. Contraspecific inhibition paired with low $\gamma_{EE}$ restores these fixed points by emphasizing competition between populations selective for opposite choices. These fixed points emerge sequentially as the inhibitory motif becomes more contraspecific: first the choice attractors appear, followed by the saddle point, and finally by the working memory attractors (arrow in Fig. 2d left). For moderate $\gamma_{EE}$, nonspecific circuits have all eight necessary fixed points, but deviations to a contraspecific motif cause the loss of the attractor for the low initial state, whereas deviations to an ipsispecific motif cause the loss of the working memory attractors, then saddle point, and then choice attractors (arrow in Fig. 2d center, Supplementary Fig. 1). For circuits with high $\gamma_{EE}$ to support decision making, inhibitory motif must be ipsispecific, as nonspecific and contraspecific circuits lack the initial low activation state attractor (Fig. 1d right, Supplementary Fig. 1). The trade-off between competition and stability across contraspecific and ipsispecific circuits is also evident in the size of choice-selective populations that support decision-making (Supplementary Fig. 2).

Specific connections between choice-selective inhibitory populations may also impact the attractors underlying decision-making. For example, competitive inhibitory-inhibitory connections can mediate disinhibition in contraspecific circuits[22,23]. We therefore investigated the effect of inhibitory-to-inhibitory connection specificity on decision-making dynamics. We extended our mean-field approach to explicitly model the activity of two choice-selective excitatory and two choice-selective inhibitory populations (Methods). This four-variable model produces the firing-rate dynamics and attractors similar to the original two-variable mean-field model (Supplementary Fig. 3a–c). In the four-variable model, we controlled the balance of connection strength between choice-selective inhibitory populations in the same manner as for other connection classes using a specificity parameter $\gamma_{II}$, which like $\gamma_{IE}$ ranges between $[-1, 1]$. We sampled the four-

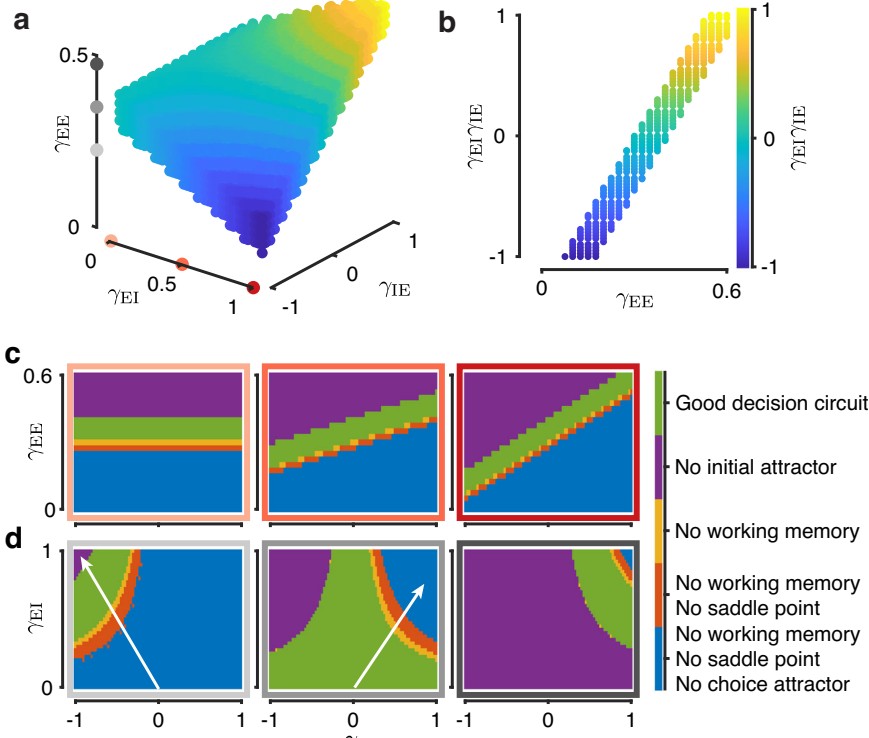

**Fig. 2 | Choice-selective inhibition expands the space of circuits supporting decision making. a** The volume in the connection specificity parameter space where circuits have all fixed points necessary to support decision-making. Color indicates the inhibitory specificity index $\gamma_{EI}\gamma_{IE}$. **b** For nonselective or nonspecific inhibition, circuits that support decision-making exist only for a narrow range of $\gamma_{EE}$. If inhibition is selective and specific, a broader range of $\gamma_{EE}$ becomes possible. Inhibitory specificity needs to be complementary to excitatory specificity, as reflected in the correlation between the inhibitory specificity index and $\gamma_{EE}$ for circuits with all necessary fixed points. **c** With increasing inhibitory selectivity, a broader range of $\gamma_{EE}$ can support decision making, evident as a steeper slope of the region with all required fixed points (green, good decision circuit). Panels show slices through the parameter space in a at $\gamma_{EI} = 0.0$ (left), $\gamma_{EI} = 0.5$ (center), $\gamma_{EI} = 1.0$ (right). Colored frames correspond to dots on the $\gamma_{EI}$ axis in **a**. **d** When $\gamma_{EE}$ is low inhibition must be contraspecific (left) and when $\gamma_{EE}$ is high inhibition must be ipsispecific (right). Slices through the parameter space in a at $\gamma_{EE} = 0.225$ (left), $\gamma_{EE} = 0.35$ (center), $\gamma_{EE} = 0.475$ (right). Colored frames correspond to dots on the $\gamma_{EE}$ axis in **a**. Arrows indicate a sequential loss of fixed points described in the text.

dimensional specificity parameter space to identify points where the eight decision-making attractors are present (Supplementary Fig. 3d). As with the two-variable model, the main factor determining whether a circuit has the necessary fixed points is the linear relationship between $\gamma_{EE}$ and $\gamma_{EI}\gamma_{IE}$. Inhibitory-to-inhibitory connection specificity $\gamma_{II}$ has a limited impact on the presence of the fixed points (Supplementary Fig. 3d, cf. Fig. 2b). Circuits with negative $\gamma_{II}$ are hyper-competitive and lose the low activation state attractor.

**Inhibitory motif controls the speed versus accuracy trade-off**

The roles enhanced by contra- and ipsispecific inhibititory motifs lead to differences in performance of decision circuits. In circuits with moderate strengths of recurrent excitation, all three motifs can support decision making for the same $\gamma_{EE}$. We found that circuits with three inhibitory motifs differ in choice accuracy on difficult trials where stimulus strength is weak (Fig. 3a). Relative to a circuit with nonspecific inhibitory outputs ($\gamma_{IE} = 0$), ipsispecific circuits are more accurate at classifying difficult stimuli but more often fail to separate the outputs sufficiently producing invalid trials (Fig. 3b). Contraspecific circuits, on the other hand, have lower accuracy for difficult stimuli. In addition, contraspecific circuits have a stimulus independent rate of trial failure attributable to trials where the firing rates of choice-selective populations separate prior to the stimulus onset (Fig. 3b), highlighting how these circuits are primed for competitive dynamics. It is well known that decision accuracy and decision time are linked through the speed-accuracy trade-off, where longer integration times lead to more accurate decisions[24–26].

Ipsispecific circuits could be more accurate at the expense of speed, so we compared the average time it takes circuits to cross the decision threshold for each stimulus strength as a proxy for decision time. Ipsispecific circuits do indeed arrive at choices more slowly than the less accurate contraspecific circuits (Fig. 3c). These differences in behavioral performance indicate a speed versus accuracy trade-off which is mediated by the specificity of connections between choice-selective populations in the circuit (also evident in the four-variable model, Supplementary Fig. 3e). These performance outcomes again highlight the roles enhanced by ipsispecific and contraspecific inhibition: the contraspecific motif primes a circuit for competition, whereas the ipsispecific motif promotes stability, lengthening integration times.

We can understand the speed-accuracy trade-off between ipsi- and contraspecific circuits by analyzing the dynamics around the saddle point. Differences in these dynamics are seen by comparing single-trial trajectories of ipsi-, non-, and contraspecific circuits in response to the neutral stimulus (Fig. 3d). At the trial start, both choice-selective populations are symmetrically activated and the trajectory moves along the stable manifold toward the saddle point. The circuit activity deviates to a choice attractor after approaching the saddle. Contra- and ipsispecific circuits differ in both how far along the stable manifold the activity progresses and how quickly it moves toward the choice attractor once it deviates. We can estimate how quickly the dynamics will leave the neighborhood of the saddle point with the time-constant $\tau_{slow}$, which is the time-constant of dynamics moving along the unstable manifold of the saddle point[4,27]. Changing

the circuit motif from contraspecific to ipsispecific by increasing $\gamma_{EI}\gamma_{IE}$ leads to an increase in $\tau_{slow}$ (Fig. 3e) and slowing down the pace of decisions (Fig. 3f). The divergence of $\tau_{slow}$ indicates that ipsispecific inhibition stabilizes the saddle point until at high $\gamma_{EI}\gamma_{IE}$ a bifurcation occurs and the saddle point becomes an attractor with a symmetric high activity state (Fig. 3g). This bifurcation leads to the system stabilizing in a state where the firing rates of two choice-selective populations do not sufficiently separate on neutral and difficult stimuli trials, a state where the circuit fails to produce a decision. Easy stimuli impose a stronger asymmetry on the phase plane[4] allowing circuits with highly ipsispecific inhibition to converge to a choice on easy trials (Supplementary Fig. 4).

### Strong ipsispecific inhibition destabilizes working memory

The inhibitory connectivity motif affects the circuit's ability to maintain the working memory of a choice. Contraspecific and nonspecific circuits maintain a difference in excitatory firing rates of at least 15 Hz for a very long time following stimulus offset, whereas ipsispecific circuits exhibit a degradation of the choice readout (Fig. 4a). This behavior can be linked to the phase plane of the unstimulated circuit. Working memory is supported by two choice attractors that are separated by saddle points from the attractor with symmetric low activity state. The separation between the working memory attractors and the saddle points is smaller for more ipsispecific circuits (Fig. 4b). For highly ipsispecific circuits, working memory attractors are extinguished after merging with the saddle points (Fig. 4b).

### Inhibitory choice selectivity in trained recurrent neural networks

So far, we used the mean-field approach to establish that choice-selective inhibition supports the function of decision-making circuits by enhancing a competitive or stabilizing role. Next, we wanted to test whether this result holds broadly by using another class of decision-making network models. We therefore trained excitatory-inhibitory recurrent neural networks (RNNs) to perform a decision-making task[28] and then tested whether inhibitory choice-selectivity regularly emerges in these networks after training and whether the dependence

between the excitatory and inhibitory specificity aligns with the two roles for inhibition. We used RNNs with 100 excitatory and 25 inhibitory units (Fig. 5a), but our results are not specific to this number of units and hold in RNNs with twice the size (Supplementary Fig. 5). Two input streams projected to all excitatory units through input weights. Two output variables were calculated as a weighted sum of excitatory unit activity. We trained RNNs to perform an identical decision-making task as the mean-field circuits by raising an output variable which corresponds to the input stream with a higher mean value. Networks were trained by back-propagation through time to minimize the mean squared error between the network outputs and predefined targets. For a given trial, a choice was recorded when the output variables became separated by a fixed threshold set to 0.25. Trials were considered invalid if the outputs separated prior to the stimulus, failed to maintain separation after stimulus offset, or separation was never achieved. We trained networks until the correct choice was made on 85% of all trials (including correct, error, and invalid trials) in a 200 trial epoch. One hundred and fifty networks reached this training threshold in 104, 343 ± 9, 264 (mean ± s.d.) trials, ranging from 83,200 to 127,600 (Fig. 5b). Networks performed the task well, making errors and failing to complete trials only for difficult stimuli (Fig. 5c). Trained networks also took longer to make decisions when presented with a difficult stimulus, similarly to mean-field circuits (Supplementary Fig. 6).

We determined whether inhibitory neurons in these RNNs were choice selective. We classified recurrent units as choice selective using receiver operator characteristic (ROC) analysis[21] (Methods). We constructed ROC curves by decoding network choice from a unit's activity on the time-step following stimulus offset. To identify which units significantly modulated their firing rate to reflect choice, we compared the area under the ROC curve ($AUC_{ROC}$) to a shuffle distribution generated from randomized trial labels (two-sided permutation test, $p < 0.05$, 150 permutations). Units that were identified as choice selective increased activation following the onset of a stimulus corresponding to their preferred choice (Fig. 5d). Inhibitory units had overall higher choice selectivity than excitatory units, as measured by the selectivity index $|AUC_{ROC} - 0.5|$ that can range from 0 to 0.5 (Fig. 5e, inhibitory 0.23 ± 0.17, excitatory 0.12 ± 0.16;

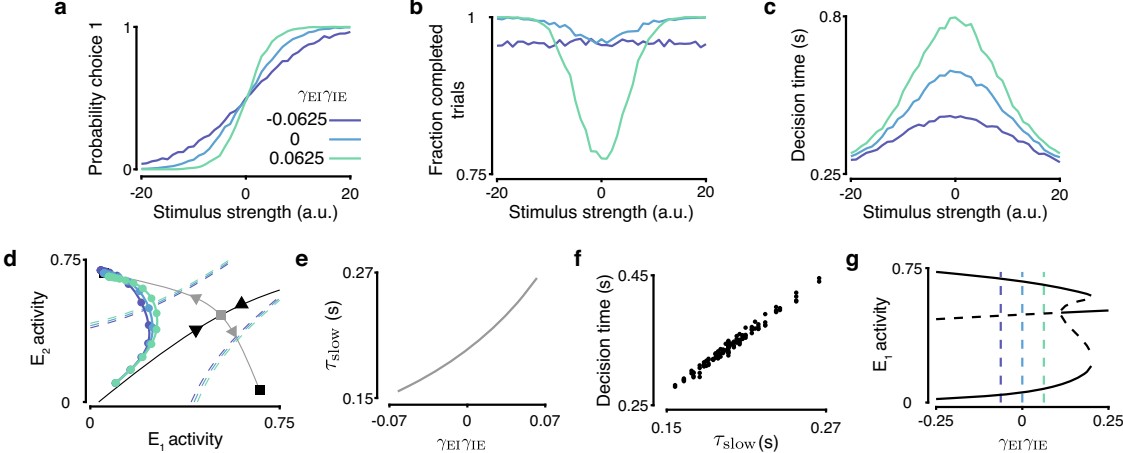

**Fig. 3 | Inhibitory circuit motifs mediate the speed-accuracy trade-off in decision-making. a–c** Contraspecific circuits are faster and less accurate, whereas ipsispecific circuits are slower and more accurate than nonspecific circuits. Psychometric functions (**a**), probability of trial completion (**b**), and chronometric functions (**c**) for circuits with different inhibitory motifs. **d** Contraspecific circuits deviate to a choice attractor earlier and faster than ipsispecific circuits. Single-trial trajectories are shown for three circuits with different inhibitory motifs, in 50 ms time steps (dots) for 0 stimulus strength. Noise is reduced by 50% for illustration clarity. The decision threshold for each circuit is shown by the dashed line. Squares indicate choice attractors, triangle indicates the saddle point. The stable (black)

and unstable (gray) manifolds of the saddle point are shown. **e** As the circuit motif changes from contra- to ipsispecific, the time constant of the unstable eigenvector of the saddle point $\tau_{slow}$ increases, indicating stabilization of dynamics and longer integration times. **f** The time constant $\tau_{slow}$ is tightly correlated with decision time (shown for stimulus strength equal to 0). **g** The saddle point becomes an attractor for ipsispecific circuits with high $\gamma_{EI}\gamma_{IE}$. The bifurcation diagram for circuits driven by a stimulus of 0 strength shows the location of attractors (black solid line) and saddle points (black dashed line). Dashed vertical lines correspond to examples in **a–d**. In all panels $\gamma_{EE} = 0.32$ and $\gamma_{EI} = 0.25$.

mean ± s.d.; Wilcoxon rank-sum test $p < 10^{-10}$). Also, the proportion of significantly selective units was higher for inhibitory than excitatory units (Fig. 5f, inhibitory 0.87 ± 0.07, excitatory 0.72 ± 0.06; mean ± s.d.; Wilcoxon Rank-Sum test $p < 10^{-10}$). Thus, inhibitory unit activity contained overall more choice information than excitatory unit activity despite the fact that only excitatory units received stimulus input. In this respect RNNs differ from experimental data in which excitatory and inhibitory neurons contained similar choice information[21].

## Excitatory specificity aligns with ispi- and contraspecific inhibitory motifs in RNNs

Based on our mean-field model, we know that for choice-selective inhibition to impact circuit function, the connections from inhibitory to excitatory populations must be specific. Therefore, after identifying choice-selective units in RNNs, we sought to determine whether the connection specificity of excitatory-excitatory and excitatory-inhibitory pairs followed the relationship predicted by the mean-field model (Fig. 2b). To analyze the specificity of connections between choice-selective populations in the RNNs, we estimated the specificity parameter $\gamma$ from the weights of trained RNNs defined in the same way as for the mean-field model (Methods). Trained networks consistently had strong excitatory-excitatory ($\gamma_{EE} = 0.59 \pm 0.07$) and excitatory-inhibitory ($\gamma_{EI} = 0.39 \pm 0.06$) specificity (Fig. 5g). This result is consistent with the constraint that inhibitory units inherit stimulus information from excitatory units to be choice or stimulus selective. Inhibitory-excitatory connections were nonspecific on average ($\gamma_{IE} = 3.6 \times 10^{-3} \pm 0.03$) but their distribution showed both ipsispecific and contraspecific motifs. Inhibitory-inhibitory connections were nonspecific on average with higher variation than inhibitory-excitatory connections ($\gamma_{II} = -5.0 \times 10^{-3} \pm 0.06$). Confirming the trend predicted by the mean-field model, excitatory specificity $\gamma_{EE}$ was correlated with the inhibitory specificity index $\gamma_{EI}\gamma_{IE}$, where networks with stronger recurrent excitation were ipsispecific and networks with weaker recurrent excitation were contraspecific (Pearson's $r = 0.53$, $p < 10^{-10}$; Fig. 5h). When comparing the connection classes individually, we found positive correlations between excitatory-excitatory, excitatory-inhibitory, and inhibitory-excitatory specificity (Fig. 5i). Inhibitory-inhibitory connection specificity was not significantly correlated with any other connection class. The higher variance and negligible correlation with other connection classes suggest that the specificity of inhibitory-inhibitory connections was unconstrained in these networks, in line with the mean-field model, where specificity of inhibitory-inhibitory connections also had a small effect on whether circuits could perform decisions (Supplementary Fig. 3d). These results show that RNNs utilize choice selective inhibition to compensate for variation in excitatory-excitatory specificity.

To further test the relationship between the excitatory and inhibitory specificity, we trained additional sets of RNNs with higher or lower excitability of excitatory units. In the mean-field model, lower (higher) excitatory gain can be compensated by either an increase (decrease) in excitatory connection specificity or by strengthening of the contraspecific (ipsispecific) motif. Accordingly, we expect that changing the activation function slope of the excitatory units in RNNs should either shift the excitatory-excitatory specificity against the direction of the gain change or shift the inhibitory specificity towards contraselective (for lower slope) or ipsielective motif (for higher slope). We trained two additional sets of networks with hypoexcitable (slope 0.5) or hyperexcitable (slope 1.5) excitatory units. Changing the excitability of excitatory units led to large shifts in $\gamma_{EE}$ without changing the distribution of inhibitory specificity (Supplementary Fig. 7). In these networks, $\gamma_{EE}$ and $\gamma_{EI}\gamma_{IE}$ were still correlated, with higher $\gamma_{EE}$ leading to higher $\gamma_{EI}\gamma_{IE}$ (Supplementary Fig. 8). These results indicate that excitatory-excitatory specificity is a higher leverage parameter that RNNs use as the most effective path to compensate for changes in

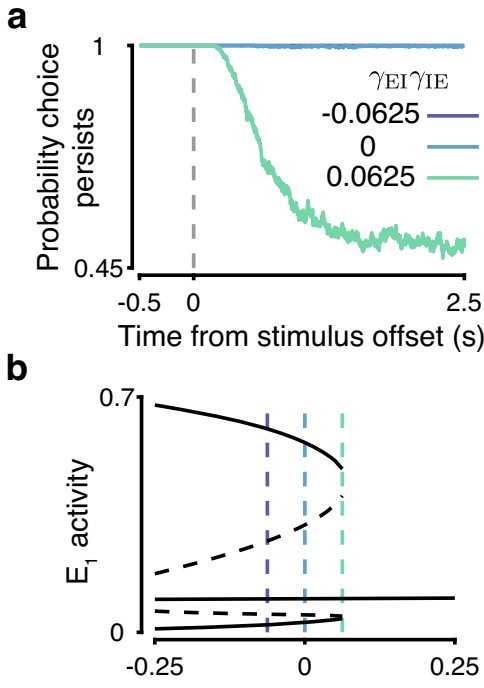

**Fig. 4 | Strong ipsispecific inhibition destabilizes working memory attractors. a** The probability of maintaining a choice after stimulus offset is diminished in ipsispecific circuits. **b** For ipsispecific circuits with high $\gamma_{EI}\gamma_{IE}$, working memory attractors are extinguished after merging with saddle points. The bifurcation diagram is for the same circuits as in Fig. 3g but in the absence of stimulus.

the excitability of excitatory units. This observation is consistent with the effect of changes in $\gamma_{EE}$ on the dynamics in the mean-field model. For accuracy, decision-time and $\tau_{slow}$, changes in $\gamma_{EE}$ are far more effective than changes in inhibitory specificity (Supplementary Fig. 9) when all other parameters are held constant. In both the mean-field and RNN models, excitatory-excitatory specificity has a larger effect than inhibitory specificity and is the main lever circuits use to compensate for changes in neural parameters.

## Perturbing inhibitory neuron activity reveals regimes where stabilizing and competitive inhibition dominate

Using the mean-field and RNN models, we established how contra- and ipsispecific inhibitory motifs enhance two different roles for inhibition in decision making circuits. To further probe these roles, we next considered how circuits respond to perturbations of inhibitory neuron activity. We used perturbations that equally targeted all inhibitory neurons irrespective of their choice selectivity by driving them with a nonspecific input $\Delta v_{0,I}$ (Fig. 6a). Such perturbations could be realized in optogenetic experiments. In circuits where the competitive role of inhibition dominates, we expect that enhancing inhibitory activity should speed up dynamics whereas suppressing inhibition should slow them down (Fig. 6b). Vice versa, in circuits where the stabilizing role of inhibition dominates, we expect that enhancing inhibitory activity should slow dynamics down and suppressing inhibition should speed them up (Fig. 6b). Because $\tau_{slow}$ provides a readily available estimate of the pace of dynamics in the mean-field model, we calculated $\tau_{slow}$ for varying nonspecific baseline input to inhibitory neurons $v_{0,I}$. We found that depending on the baseline level of inhibitory activity both regimes are possible in the mean-field circuit: one where competitive role dominates and one where stabilizing role dominates (Fig. 6c). Around a low baseline value of inhibitory activity ($v_{0,I} = 11.5$ in Fig. 6c), contra-, ipsi-, and nonspecific circuits respond to perturbations similarly, such that enhancing inhibition ($\Delta v_{0,I} > 0$) leads to a decrease in $\tau_{slow}$, i.e.

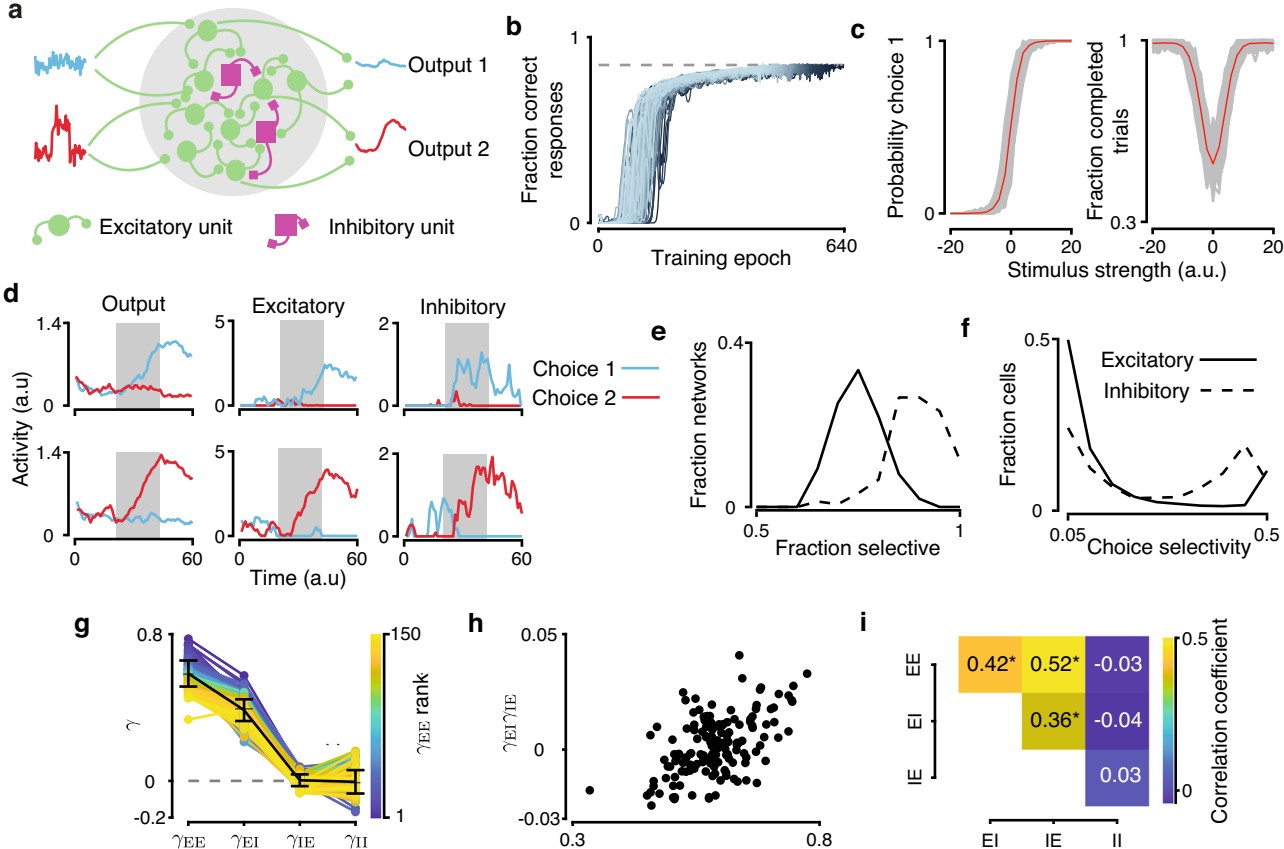

**Fig. 5 | Inhibitory choice selectivity in trained recurrent neural networks.**
**a** RNNs are trained to compare two inputs and indicate which has higher mean by elevating the corresponding output. RNNs are composed of 100 excitatory and 25 inhibitory units. **b** We trained 150 RNNs to a consistent level of performance. RNN performance improved gradually during training. We stopped the training when the network performance reached 85% correct responses (gray dashed line). Lines show individual RNNs, color gradient indicates the network's rank to reach 85% performance. **c** Psychometric functions (left) and probability of trial completion (right) for all trained RNNs (gray) and the mean across 150 networks (red). **d** Excitatory and inhibitory units in trained RNNs display choice selectivity. Traces show the activity of the RNN outputs (left), two excitatory units (center), and two inhibitory units (right) on two example trials with choice 1 (upper row, stimulus strength −20) and choice 2 (lower row, stimulus strength 20). Gray shading

indicates the stimulus period. **e** In trained RNNs, the overall choice selectivity is greater for inhibitory than excitatory units. Distributions show the choice-selectivity index across all units from all networks. **f** In trained RNNs, the fraction of units with significant choice selectivity is greater for inhibitory than excitatory units. Distributions show the fraction of selective units across networks. **g** Trained RNNs show a range of specificity parameters $\gamma$ for each connection class (colored lines - individual RNNs, black - mean ± s.d. across 150 networks). Color indicates networks sorted by $\gamma_{EE}$. **h** In trained RNNs, the specificity index $\gamma_{EI}\gamma_{IE}$ is positively correlated with $\gamma_{EE}$. **i** In trained RNNs, excitatory-excitatory, excitatory-inhibitory, and inhibitory-excitatory specificity are correlated, whereas inhibitory-inhibitory specificity is uncorrelated with other connection classes. * indicates significant correlation ($p < 0.05$, two-tailed permutation test, 5000 permutations).

faster dynamics. Around a high baseline value of inhibitory activity ($v_{0,I} = 14$ in Fig. 6c), all circuits respond in the opposite way, such that enhancing inhibition increases $\tau_{slow}$. This U-shaped dependence of $\tau_{slow}$ on the baseline input to inhibitory neurons $v_{0,I}$ results from the system approaching bifurcation points at either extreme of the parameter range that supports decision making[4] (Supplementary Fig. 10). These two regimes–a low inhibition and a high inhibition regime–differ in which role of inhibition dominates: competitive or stabilizing, respectively. The inhibitory motif (contra-, non-, or ipsispecific) further shifts this emphasis within the constraints of each regime. These regimes can be identified via perturbations by characterizing how the circuit dynamics respond to changes in inhibitory tone.

To confirm the existence of competitive and stabilizing regimes, we perturbed the mean-field circuits around the low and high baseline values of the inhibitory activity. We enhanced or suppressed inhibition during the stimulus period of a trial and measured changes in the circuit performance. We constructed a set of metrics to quantify changes in the fraction of completed trials, decision time, and choice accuracy relative to the unperturbed circuit for all stimulus strengths. The effects of these perturbations followed the predictions from the

calculation of $\tau_{slow}$ (Fig. 6d–k). Enhancing inhibition decreased decision time in the low inhibition regime, but increased decision time in the high inhibition regime (cf. Fig. 6d, f and h, j). Consistent with the slowing effects of the perturbation, circuits in the high inhibition regime failed more often to complete trials (Fig. 6e) and became more accurate (Fig. 6g) when inhibition was enhanced. Circuits in the low inhibition regime showed the opposite behavior (Fig. 6h–k). Thus, by perturbing inhibitory neuron activity we can determine whether the competitive or stabilizing inhibition dominates in a circuit.

We then delivered enhancing or suppressing perturbations to inhibitory units in trained RNNs during the stimulus period to identify in which inhibitory regime these networks operate. Enhancing inhibition increased decision times, reduced the fraction of completed trials, and increased accuracy, consistent with these RNNs operating in the stabilizing inhibition regime (cf. Fig. 6h–k and l–o).

## Discussion
We showed that choice selectivity of inhibitory neurons can affect the function of decision making circuits by enhancing one of two roles for inhibition: facilitating competition or stabilizing recurrent excitation. In

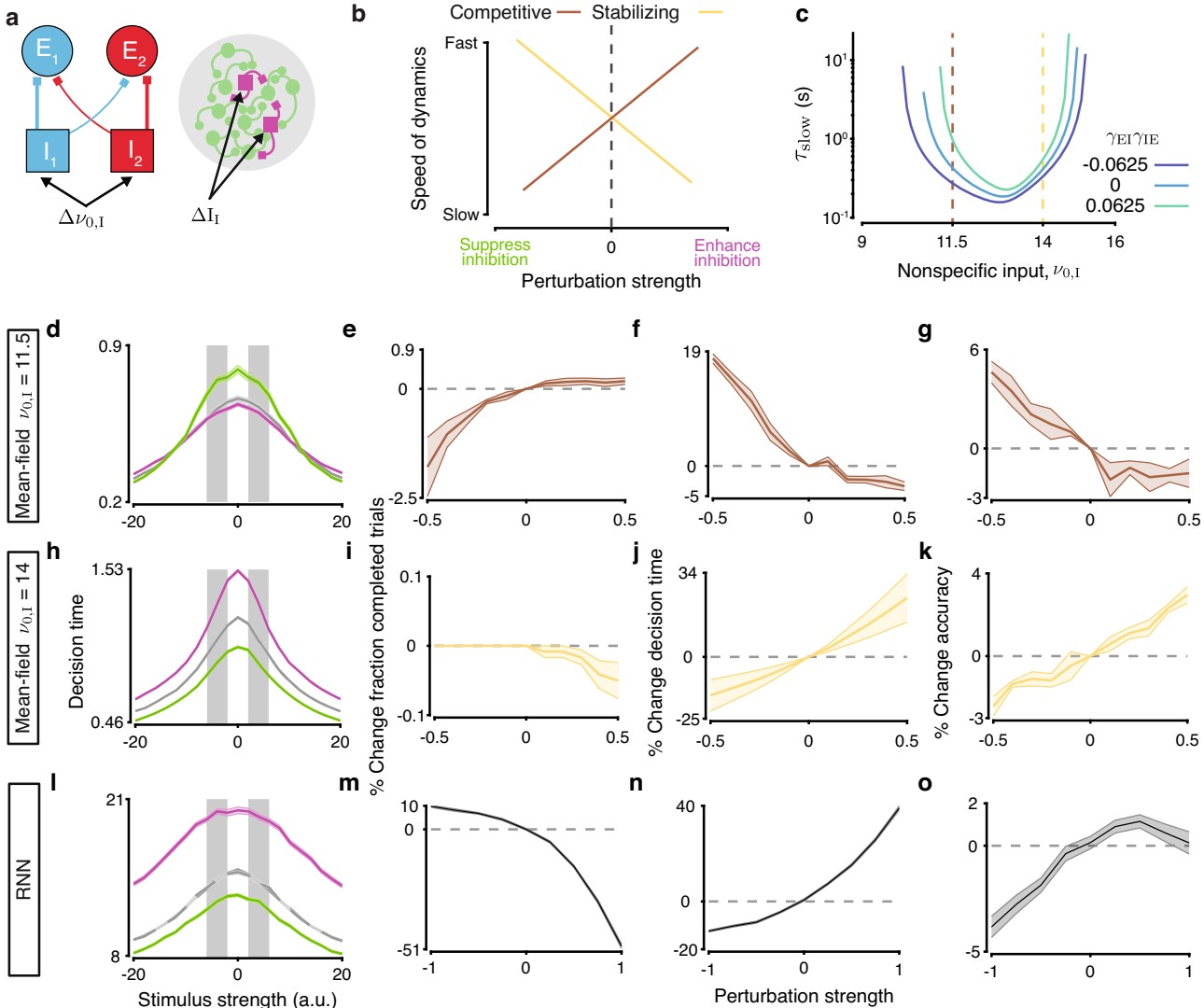

**Fig. 6 | Perturbations to inhibitory activity reveal regimes where stabilizing and competitive inhibition dominate. a** We perturbed mean-field models by delivering a nonspecific input $\Delta\nu_{0,I}$ to all inhibitory neurons during stimulus period. Perturbations were similarly delivered to RNNs by adding a constant input $\Delta I_I$ to all inhibitory units during the stimulus. **b** Circuits where competitive (brown) or stabilizing (yellow) inhibition dominates are predicted to have diverging responses to perturbations of inhibitory activity. **c** In the mean-field model, dependence of $\tau_{slow}$ on the baseline input to inhibition $\nu_{0,I}$ reveals two inhibitory regimes: a low inhibition regime where competitive inhibition dominates and a high inhibition regime where stabilizing inhibition dominates. Contra- or ipsispecific inhibitory motifs shift the emphasis within these regimes (e.g., $\tau_{slow}$ is always longer for ipsi- than contraspecific circuits). The effects of inhibitory perturbations in the mean-field model differ between these two regimes depending on the baseline value $\nu_{0,I}$.

**d**–**g** Around a low baseline ($\nu_{0,I} = 11.5$, brown line in **c**), enhancing inhibition speeds up decision times (magenta line in **d**; **f**), increases the rate of trial completion (**e**), and decreases accuracy (**g**), whereas suppressing inhibition produces the opposite effects, e.g., slows down decision times (green line in **d**; **f**). Results are shown for nonspecific circuits. Gray areas in d indicate stimulus strengths used to calculated the values in **e**–**g**. Error bars indicate ± s.e.m. across 2000 trials. **h**–**k** Same as **d**–**g** for the high inhibition regime ($\nu_{0,I} = 14$, yellow line in **c**). Perturbations of inhibitory activity produce the reversed effects. Error bars indicate ± s.e.m. across 2000 trials. **l**–**o** Same as **h**–**k** for perturbations of inhibitory neurons in RNNs. RNN's response to perturbations mirrors the effects in the mean-field model in the stabilizing regime (c.f. **h**–**k**). Enhancing inhibition in RNNs slows down decision times, decreases the rate of trial completion, and increases accuracy. Error bars indicate ± s.e.m. across 75 networks.

---

the mean-field model, choice selective inhibition and specific connections from inhibitory to excitatory populations expand the excitatory-excitatory specificity parameter space of circuits that support decision-making. For the range of excitatory connection specificities supporting both ipsispecific and contraspecific inhibitory circuits, the speed and accuracy of decisions tightly depend on whether the ipsi- or contraspecific inhibitory motif is present. Inhibitory choice selectivity also emerges in RNNs trained to perform a decision-making task, and the specificity of excitatory and inhibitory connections within trained RNNs is correlated, consistent with the mean-field model predictions. The mean-field model further predicts the existence of two dynamical regimes: (i) a low-inhibition regime where the competitive role

dominates, and (ii) a high-inhibition regime where stabilizing role dominates. In trained RNNs, perturbations of all inhibitory neurons indicate that these networks operate in the stabilizing inhibition regime.

Decision-making circuits with non-selective inhibition exist only within a narrow range of excitatory-excitatory connection specificity. When inhibitory neurons inherit choice-selectivity from excitatory neurons and also project to excitatory neurons via specific connections, a broad range of circuit configurations can support decision-making. In circuits capable of decision-making, the correlation between the specificity of excitatory ($\gamma_{EE}$) and inhibitory connections ($\gamma_{EI}\gamma_{IE}$) reveals how the contra- and ipsispecific motifs enhance one of two roles for inhibition: facilitate competition between populations

coding for opposite choices or stabilize amplification driven by strongly recurrent excitation. When $\gamma_{EE}$ is low and excitatory populations alone cannot drive selective activation, contraspecific inhibitory motifs support decision-making by maximizing competition. Conversely, when $\gamma_{EE}$ is high and excitatory self-amplification becomes unstable, ipsispecific inhibitory motifs stabilize firing rates.

The categorical output of decision-making circuits is thought to be driven by strongly selective excitatory to excitatory selectivity with the evidence accumulation based on amplification through NMDA receptors[2,4]. In these models the specificity of excitatory connections is sufficient to drive competition and selective activation. We found that deviations from a narrow range of $\gamma_{EE}$ require complementary inhibitory circuitry. When recurrent excitatory specificity is low, contraspecific inhibition is required to form the attractors needed for decision-making computation. This mechanism was described in circuits where excitatory populations have limited capacity for amplification, such as the midbrain circuit in the owl[22], and in linear integrator models[29]. On the other hand, when recurrent excitatory specificity is high, the strong excitatory feedback amplification needs matching ipsispecific inhibition to stabilize the circuit. This mode of inhibitory selectivity is known to improve stability and robustness of a circuit to perturbations[17,30]. Additionally, shifts in E/I balance through modulation of gain or synaptic efficacy can improve the robustness and parameter range of decicion-making circuit models[31,32].

We found a similar relationship between excitatory and inhibitory connection specificity in RNNs suggesting the balance between competitive and stabilizing inhibition is a general principle in E-I networks. While specific connections between excitatory and inhibitory units were clearly important for the decision-making function in our networks, connections between inhibitory units appeared unconstrained, indicating this connection class has limited effect on circuit function like in the mean-field model. RNNs are increasingly often used to develop theories of how neural circuits perform computations[23,28,33]. Some studies trained RNNs under the constraint that units have either exclusively excitatory or exclusively inhibitory outputs[28,34] (Dale's law). Studies of E-I RNNs which focus on the impact of inhibitory connections show that specificity of inhibitory-inhibitory connections can be critical to circuit function[23]. The apparent difference in the importance of inhibitory-inhibitory selectivity between our networks and previous work could result from differences in the training procedures[35]. We observed a large impact of RNN training hyperparameters on the emerging circuit structure. Future work is needed to understand how details of training influence the emerging circuit structure and computations performed by RNNs.

Our results show that selective inhibition can have a marked effect on the function of neural circuits. Many models of categorical decision-making rely on a nonspecific pool of inhibitory neurons to enforce winner-take-all competition between excitatory neurons[2,3]. While these models reproduce the dynamics of decision-making circuits they do not fully account for the diversity of interneurons within the cortex. Cortical inhibitory neurons show selective activation in many modalities including primary sensory[13,17,36,37] and association areas[19–21]. Moreover, choice-selectivity of parietal inhibitory neurons is equal to that of excitatory neurons during an audio-visual discrimination task[21].

In the mean-field model, we assume that choice selectivity of inhibitory neurons arises from specific connections from choice-selective excitatory neurons ($\gamma_{EI}$ in our model). While it is possible that choice selectivity could arise from external inputs to interneurons[38] or even from random connections between excitatory and inhibitory neurons[39], most circuit models assume stimulus information is exclusively provided by inputs to excitatory neurons. Inhibitory choice-selectivity also emerged in our RNNs trained to perform 2AFC task[28]. In our RNNs, inhibitory units can only inherit stimulus or choice information through specific connections from excitatory populations, unlike in other trained RNNs[23]. For both excitatory and inhibitory units

in trained RNNs, we found that the fraction of selective units was higher than is commonly used in circuit models[2,4] and found in experiments[21]. This difference could be due to the simplicity of RNNs compared to in vivo circuits, and also a training process which aims to minimize total activity through regularization. In addition, decisions in the RNN are fully determined by the local circuit, whereas an animal's behavioral output arises from a broadly distributed circuitry. Although higher choice selectivity for inhibitory units was robust to doubling the network size (Supplementary Fig. 5b, c), it could result from the need to leverage all of these units in a network much smaller than those in the brain.

The core computation of the model is the selective activation of a single excitatory population when the stimulus is presented and a mechanism to integrate stimulus information before diverting to a choice attractor. By enhancing stability, ipsispecific circuits lengthen the period when a circuit can maintain mutual activation of populations encoding competing choices, thus increasing the integration window which leads to more accurate stimulus classifications. Contraspecific circuits, primed for competition, minimize the integration period which increases error frequency.

In attractor networks, modulation of $\tau_{slow}$ for controlling the speed and accuracy of decisions is well known and can arise from other mechanisms than inhibitory output specificity. In the model with nonspecific inhibition, $\tau_{slow}$ increases with stimulus difficulty[4] and can be also modulated via top-down excitation[27]. Our finding that excitatory-inhibitory connectivity influences this well established mechanism highlights the importance of inhibitory circuitry to evidence accumulation. A key difference between controlling $\tau_{slow}$ via inhibitory motif versus top-down excitation is that the location of the saddle point is unaffected by $\gamma_{IE}$ whereas increasing top-down excitation shifts the saddle towards the origin, effectively acting as a collapsing decision-bound[27]. Top-down excitation can be adjusted rapidly from one trial to the next to match the decision's speed and accuracy to the task demands. Could the inhibitory motif also be dynamically changed to meet changing task requirements? Modulation of the speed-accuracy trade-off through changes of the inhibitory motif may be mediated by activation or inactivation of inhibitory subpopulations connected in either a contraspecific or ipsispecific pattern (representing a shift in $\gamma_{IE}$ for the circuit as a whole).

Selective neuromodulatory control of genetically identifiable inhibitory subtypes may provide for control of inhibitory motifs. Inhibitory subtypes have distinct connectivity patterns to neighboring excitatory neurons: fast-spiking cells have far more reciprocal connections to excitatory neurons than adapting interneurons[16]. A shift in output specificity could be mediated through top-down activation of inhibitory subnetworks or through neuromodulation of distinct inhibitory subtypes such as $PV^+$, $SOM^+$, or $VIP^+$. Acetylcholine has layer-dependent effects on the responsiveness of both regular spiking and fast spiking neurons in the visual cortex, which could differentially activate distinct inhibitory motifs on behaviorally relevant timescales[40–42]. Additionally, acetylcholine can reduce the release of inhibitory neurotransmitters in cortical neurons[43], thus directly affecting inhibitory connectivity.

Our mean-field framework reduces the dynamics of the full network with 6 excitatory and inhibitory populations to a two-variable system using several approximations, in particular, the steady-state assumption for GABA dynamics[4]. This assumption is based on the timescale separation between decay time constants of slow NMDA (~100 ms) and fast GABA (~5 ms) conductance. The slow NMDA dynamics dominate the time evolution of network activity, and one can assume that all other variables reach their steady-state nearly instantly[4]. Despite being fast, the dynamics of GABA synapses can also affect decision-making behavior[44]. A study[44] considered a set of circuits with parameters chosen so that when the steady-state assumption is applied, all models reduce to the two-variable model with the exact

same parameter set. Thus, all differences in dynamics of these circuits were driven by GABA dynamics. In these circuits, the GABA dynamics mediated a speed-accuracy trade-off and, moreover, this tradeoff was more efficient in circuits with selective inhibition[44]. While this study considered only ipsispecific inhibitory connectivity and a narrow space of circuits that all map onto a single parameter set of a two-variable model, our work explores a wide range of circuit configurations ranging from contraspecific to ipsispecific inhibitory motifs. Our findings are robust to the steady-state approximation of GABA dynamics as we show using a four-variable mean-field model (Supplementary Fig. 3). Together these results show that inhibitory connectivity motifs and GABA dynamics both affect decision-making behavior.

Another key performance metric that depends on selective inhibition is the rate of trial completion. Our models (both the mean-field and RNNs) fail to reach the imposed decision threshold on a fraction of trials with low stimulus strength, which we call invalid trials. This behavior is common across spiking[2,32,45], mean-field[4,44] and RNN[28] models of decision-making. Our treatment of invalid trials is conservative, as we report invalid trials as a separate behavioral outcome different from correct or incorrect decision[32], whereas most other studies assign a choice at random on trials when the network does not reach the decision threshold[2,4,28,44,45]. The random assignment of choices on invalid trials can conceal differences in network dynamics, making distinct dynamical regimes indistinguishable in psychometric functions[45]. We find that the completion rate of difficult trials is reduced in circuits where stability is emphasized due to increased integration time. Circuit models frequently differ from experimental subjects in the rate of trial completion, which was attributed to an urgency signal gating the evidence accumulation process which is absent in circuit models[46–49]. One possible mechanism for an urgency signal in decision circuits could be a nonspecific external ramping input[50]. Incorporating such inputs into future models of decision-making would be an important next step in the study of selective inhibition.

We show that choice selective inhibition can enhance one of two roles for inhibition in decision-making circuits: facilitating competition or stabilizing excitatory feedback. Both these roles are simultaneously fulfilled by inhibition in any decision making circuit. Enhancing activity of all inhibitory neurons can shift the circuit from a regime where the competitive role dominates to a regime where the stabilizing role dominates regardless of which inhibitory motif is present. This effect echos results which find shifts in E/I balance can induce leaky or unstable integration[45]. The stabilizing and competitive regimes can be differentiated by the behavioral response to perturbations of inhibitory activity. Perturbations during reaction time tasks should reveal which inhibitory role is dominant in vivo. The balance of these two roles is critical for circuits to perform decision tasks, and shifts in this balance could align dynamics with changing task requirements. More experimental work is needed to uncover how inhibitory subnetworks strike this balance in the cortex. Specifically, whether functional selectivity is constrained to certain inhibitory subtypes and whether inhibitory neurons are recruited to perform a task in a state dependent manner are important questions for future work.

## Methods
### Mean-field model
Our mean-field model accounts for interactions among 6 populations: 3 excitatory (2 choice selective and 1 nonselective) and 3 inhibitory (2 choice selective and 1 nonselective). Including nonselective neurons in the model is consistent with previous work[4] and reflects the experimental observation that only a fraction of all recorded neurons shows choice selectivity[21]. Each selective population contains the fraction $f$ of the total number $N_E$ ($N_I$) of excitatory (inhibitory) neurons, so that $1 - 2f$ is the proportion of nonselective neurons. We reduce the dynamics of the full network with 6 excitatory and inhibitory

populations to a dynamical system with two variables representing the activations of N-methyl-D-aspartate (NMDA) conductances (in terms of fraction of channels open) for synapses originating from two choice-selective excitatory populations[4]. The model reduction to two dimensions leverages the timescale separation between decay time constants of the slow NMDA (~100 ms) and fast $\gamma$-aminobutyric acid (GABA, ~5 ms) and $\alpha$-amino-3-hydroxy-5-methyl-4-isoxazolepropionic acid (AMPA, ~2 ms) receptors. The slow NMDA dynamics dominate the time evolution of the system, and one can assume that all other variables reach their steady-state nearly instantly[4]. The dynamics of the NMDA activation variable for population $i$ ($i \in \{1, 2\}$) are governed by:

$$\frac{dS_i}{dt} = \frac{-S_i}{\tau_{NMDA}} + (1 - S_i)\gamma\Phi(x_i), \quad (1)$$

where $\tau_{NMDA} = 0.1$ s and $\gamma = 0.641$. The non-linear function $\Phi$ transforms input current $x_i$ [nA] into firing rate:

$$\Phi(x_i) = \frac{ax_i - b}{1 - e^{-d(ax_i - b)}}, \quad (2)$$

where $a = 270$ nC$^{-1}$, $b = 108$ Hz, and $d = 0.154$ s. The input to population $i$ is:

$$x_i = \alpha_1(\gamma_{EE}, \gamma_{EI}, \gamma_{IE})S_i + \alpha_2(\gamma_{EE}, \gamma_{EI}, \gamma_{IE})S_j + I_{0,i}(\gamma_{EE}, \gamma_{EI}, \gamma_{IE}) + I_{stim,i} + I_{\eta,i}, \quad (3)$$

where index $j$ refers to the other excitatory population. The complexity of the circuit structure, including interactions between all selective and nonselective excitatory and inhibitory neurons, is collapsed into two-dimensional model through the variables $\alpha_1$, $\alpha_2$, and $I_{0,i}$ as described in the section Circuit Structure below.

The stimulus $I_{stim,i}$ is defined as an increase in the rate of external excitatory inputs to choice-selective excitatory neurons of magnitude $\mu$. We define the strength of evidence for one versus the other choice as stimulus coherence $c$, which can range between $-100\%$ and $100\%$. For population $i$ the stimulus is then defined as:

$$I_{stim,i}(t,\mu,c) = \begin{cases} J_{AMPA,ext}\mu(1 - \frac{c}{100}) & t_{stim,on} < t < t_{stim,off}, i = 1, \\ J_{AMPA,ext}\mu(1 + \frac{c}{100}) & t_{stim,on} < t < t_{stim,off}, i = 2, \\ 0 & \text{otherwise}. \end{cases} \quad (4)$$

For all cases, we set $\mu$ to 40 Hz. Noise is introduced through the inputs $I_{\eta,i}$ to the two excitatory populations filtered through fast synaptic activation of AMPA receptors:

$$\frac{dI_{\eta,i}}{dt} = -\frac{I_{\eta,i}}{\tau_{AMPA}} + \frac{\eta(t)}{\sqrt{\tau_{AMPA}}}, \quad (5)$$

where $\tau_{AMPA}$ is 0.002 s and $\eta(t)$ is a white Gaussian noise with zero mean and standard deviation 0.02 nA. We performed numerical simulations using the Euler method with a 2 ms time step.

### Circuit structure
We derived two-dimensional mean-field equations, which model the dynamics of the entire circuit through the effective interaction strengths $\alpha_1$, $\alpha_2$ between the two excitatory populations, and the background currents $I_{0,i}$. This reduced model is based on approximating the firing rates of all three inhibitory populations (two choice-selective and one nonselective) and of the nonselective excitatory population as linear functions of their inputs. Thus, the firing rates of these populations change linearly in response to changes in the firing rates of the two explicitly modeled excitatory populations E$_1$ and E$_2$[4]. We define $\alpha_1$ as a term which describes how activity $S_{1(2)}$ from the excitatory population E$_{1(2)}$ filters through the circuit (i.e. via E$_{2(1)}$, E$_0$, I$_0$,

$I_1$, $I_2$, and feeding back onto itself) to impact its own firing rate. Similarly, $\alpha_2$ describes how the activity $S_{1(2)}$ filters through the circuit to impact the firing rate of the opposite excitatory population. $I_{0,i}$ describes the net input from the population activity that does not depend on the activity of $E_1$ or $E_2$. Thus, this model accounts for interactions between all six populations with only two dynamical system equations Eq. (1).

We parametrized connection specificity between choice-selective populations by $\gamma_{JK}$ between presynaptic population $J$ and postsynaptic population $K$. The index $J, K \in \{E, I\}$ defines neuron type as excitatory or inhibitory. We translate $\gamma_{JK}$ to a synaptic weight under a constraint that the total input to each population remains constant for all values of $\gamma_{JK}$. To this end, we defined an intermediate weight $\hat{w}_{JK} = N_s w_J / (N_s + \gamma_{JK}(2 - N_s))$, where $N_s = 2$ is the number of competing choice-selective populations and $w_E = w_I = 1$. We then set connection weights between populations with the same choice selectivity to $w_{JK}^+ = \hat{w}_{JK} + \gamma_{JK}\hat{w}_{JK}$ and between populations with opposite selectivity to $w_{JK}^- = \hat{w}_{JK} - \gamma_{JK}\hat{w}_{JK}$. We can rewrite $\gamma$ in terms of $w^+$ and $w^-$ as:

$$\gamma = \frac{w^+ - w^-}{w^+ + w^-}. \tag{6}$$

Connections to and from nonselective neurons were held at $w_J = 1$. This definition enforces that all neurons receive the same total input weight for any value of $\gamma_{JK}$. We set the specificity parameter $\gamma_{EE} = 0.32$ as in refs. [2,4], except in Figs. 1 and 2. We set $\gamma_{EI} = 0.25$ except in Figs. 1 and 2.

The effective interaction strengths $\alpha_1$ describes the recurrent feedback from an excitatory population's activity onto itself fed through other populations in the circuit. This term consists of four components $\alpha_1 = \lambda_1(\alpha_{1a} + \alpha_{1b} + \alpha_{1c} + \alpha_{1d})$:

$$\alpha_{1a} = f N_E w_{EE}^+ J_{NMDA,eff,E}, \tag{7}$$

$$\alpha_{1b} = \frac{1}{\kappa g_{I2}} (c_I f N_E w_{EI}^+ J_{NMDA,eff,I})(f w_{IE}^+ N_I J_{GABA,E} \tau_{GABA}), \tag{8}$$

$$\alpha_{1c} = \frac{1}{\kappa g_{I2}} (c_I f N_E w_{EI}^- J_{NMDA,eff,I})(f w_{IE}^- N_I J_{GABA,E} \tau_{GABA}), \tag{9}$$

$$\alpha_{1d} = \frac{1}{\kappa g_{I2}} (c_I f N_E w_E J_{NMDA,eff,I})(f w_I N_I J_{GABA,E} \tau_{GABA}). \tag{10}$$

These components of $\alpha_1$ account for the effect of an excitatory population's activity on its own activity filtered via (a) direct self-coupling, (b) the activity of the inhibitory population with the same choice selectivity, (c) the activity of the inhibitory population with the opposite choice selectivity, and (d) the activity of nonselective inhibitory neurons. Similarly, $\alpha_2$ describes the influence of one excitatory population's activity onto the other fed through all other populations in the circuit and also consists of four components $\alpha_2 = \lambda_2(\alpha_{2a} + \alpha_{2b} + \alpha_{2c} + \alpha_{2d})$:

$$\alpha_{2a} = f N_E w_{EE}^- J_{NMDA,eff,E}, \tag{11}$$

$$\alpha_{2b} = \frac{1}{\kappa g_{I2}} (c_I f N_E w_{EI}^- J_{NMDA,eff,I})(f w_{IE}^+ N_I J_{GABA,E} \tau_{GABA}), \tag{12}$$

$$\alpha_{2c} = \frac{1}{\kappa g_{I2}} (c_I f N_E w_{EI}^+ J_{NMDA,eff,I})(f w_{IE}^- N_I J_{GABA,E} \tau_{GABA}), \tag{13}$$

$$\alpha_{2d} = \frac{1}{\kappa g_{I2}} (c_I f N_E w_E J_{NMDA,eff,I})(f w_I N_I J_{GABA,E} \tau_{GABA}). \tag{14}$$

The components of $\alpha_2$ account for the effect on an excitatory population's activity from the oppositely selective excitatory population's activity filtered via (a) direct coupling, (b) the activity of the inhibitory population with the same selectivity, (c) the activity of the inhibitory population with the opposite selectivity, and (d) the activity of nonselective inhibitory neurons. The effects of nonselective neurons and external background inputs are described by $I_{0,i} = \lambda_I(I_{0,ia} + I_{0,ib} + I_{0,ic} + I_{0,id})$:

$$I_{0,ia} = (1 - N_s f) N_E w_E J_{NMDA,eff,E} \psi_{3,in}, \tag{15}$$

$$I_{0,ib} = I_{AMPA,ext,i} - (1 - N_s f) w_I N_I J_{GABA,E} \tau_{GABA}(\nu_{0,I} + (c_I I_{0,I} - I_{m,I})/g_{I2})/\kappa, \tag{16}$$

$$I_{0,ic} = -f w_{IE}^+ N_I J_{GABA,E} \tau_{GABA}(\nu_{0,I} + (c_I I_{0,I} - I_{m,I})/g_{I2})/\kappa, \tag{17}$$

$$I_{0,id} = -f w_{IE}^- N_I J_{GABA,E} \tau_{GABA}(\nu_{0,I} + (c_I I_{0,I} - I_{m,I})/g_{I2})/\kappa, \tag{18}$$

where:

$$I_{AMPA,ext,i} = J_{AMPA,ext,E} \tau_{AMPA} N_{ext} \nu_{ext}, \tag{19}$$

$$I_{0,I} = I_{AMPA,ext,I} + J_{NMDA,eff,I} w_E (1 - N_s f) N_E \psi_{3,in}, \tag{20}$$

$$I_{AMPA,ext,I} = J_{AMPA,ext,I} \tau_{AMPA} \nu_{ext}, \tag{21}$$

$$\psi_{3,in} = \frac{\gamma \tau_{NMDA} \nu_{3,in}}{1 + \gamma \tau_{NMDA} \nu_{3,in}}. \tag{22}$$

These terms account for the input to the excitatory population $E_i$ from the nonselective excitatory population filtered via (a) direct coupling, (b) the nonselective inhibitory population, (c) the inhibitory population with the same choice selectivity, (d) the inhibitory population with the opposite selectivity. The term $\psi$ accounts for the NMDA activation of nonselective excitatory neurons. We calculated the firing rate of inhibitory populations as $\Phi_{I,1(2)} = \alpha_{1,I} S_{1(2)} + \alpha_{2,I} S_{2(1)} + I_{0,II}$, where:

$$\alpha_{1,I} = (c_I f N_E w_{EI}^+ J_{NMDAeff,I})/g_{I2}, \tag{23}$$

$$\alpha_{2,I} = (c_I f N_E w_{EI}^- J_{NMDAeff,I})/g_{I2}, \tag{24}$$

$$I_{0,II} = \nu_{0,I} + (c_I I_{0,I} - I_{m,I})/g_{I2}. \tag{25}$$

All parameter values are provided in Table 1.

## Evaluation of circuit performance

We considered a trial to be valid if the following criteria were met: (i) the firing rate difference between the two choice selective excitatory populations was less than 5 Hz for the entire period prior to stimulus onset, (ii) the firing rate difference was above the decision threshold of 15 Hz for at least one time step during the stimulus period and the time point following stimulus offset. Fraction completed trials for each stimulus level was defined as the number of valid trials out of all trials presented. Only valid trials were considered for computing chronometric and psychometric functions. Our treatment of invalid trials is more conservative than in many other studies, as we report invalid

## Table 1 | Mean-field model parameters

| Parameter | Value | Description |
|---|---|---|
| **Mean-field physiological constants** | | |
| $N_E$ | 1600 | Number of excitatory neurons |
| $N_I$ | 400 | Number of inhibitory neurons |
| $N_{ext}$ | 800 | Number of external inputs |
| $N_s$ | 2 | Number of possible choices/ choice selective populations |
| $f$ | $\in [0.13, 0.2]$ | Size of each selective population as a fraction of all E or I neurons |
| $\tau_{NMDA}$ (s) | 0.1 | Slow excitatory synaptic time constant |
| $\tau_{AMPA}$ (s) | 0.002 | Fast excitatory synaptic time constant |
| $\tau_{GABA}$ (s) | 0.005 | Inhibitory synaptic time constant |
| $\gamma$ | 0.641 | Firing-rate to NMDA activation scaling factor |
| $I_{ml}$ (Hz) | 177 | Inhibitory f-I curve intercept |
| $c_I$ (Hz/nA) | 615 | Inhibitory f-I curve slope |
| $g_{I2}$ | 2 | Inhibitory f-I curve scaling factor |
| $v_{0,I}$ (Hz) | $\in [9, 16]$ | Rate of background input to inhibitory neurons |
| $v_{ext}$ (Hz) | 3 | Rate of background input to selective excitatory neurons |
| $v_{3,in}$ (Hz) | 2 | Rate of background input to non-selective excitatory neurons |
| $V_E$ (mV) | −53.4 | Excitatory neuron resting potential |
| $V_I$ (mV) | −52.1 | Inhibitory neuron resting potential |
| $E_E$ (mV) | 0.0 | Excitatory synapse reversal potential |
| $E_I$ (mV) | −70.0 | Inhibitory synapse reversal potential |
| $g_{E,rec,NMDA}$ ($\mu$S) | $1.95 \times 10^{-4}$ | Maximum recurrent NMDA conductance, excitatory neurons |
| $g_{I,rec,NMDA}$ ($\mu$S) | $1.02 \times 10^{-4}$ | Maximum recurrent NMDA conductance, inhibitory neurons |
| $g_{E,rec,GABA}$ ($\mu$S) | 0.130 | Maximum recurrent GABA conductance, excitatory neurons |
| $g_{I,rec,GABA}$ ($\mu$S) | 0.0084 | Maximum recurrent GABA conductance, inhibitory neurons |
| $g_{E,ext,AMPA}$ ($\mu$S) | $2.1 \times 10^{-3}$ | Maximum external AMPA conductance, excitatory neurons |
| $g_{I,ext,AMPA}$ ($\mu$S) | $1.62 \times 10^{-3}$ | Maximum external AMPA conductance, inhibitory neurons |
| $J_{AMPA,ext}$ (nA/Hz) | $5.2 \times 10^{-4}$ | External stimulus current due to a single input event |
| $\lambda_1$ | 1.6719 | Scaling factor for $\alpha_1$ |
| $\lambda_2$ | 1.8844 | Scaling factor for $\alpha_2$ |
| $\lambda_I$ | 0.9229 | Scaling factor for $I_{0,i}$ |
| **Mean-field derived constants** | | |
| $\kappa$ | $1 + \frac{c_I}{g_{I2}} N_I J_{GABA,I} \tau_{GABA}$ | Linearized factor for inhibitory neurons |
| $J_{GABA,E}$ (nA) | $-g_{E,rec,GABA}(E_I - V_E)$ | Effective GABA current, excitatory neurons |
| $J_{GABA,I}$ (nA) | $-g_{I,rec,GABA}(E_I - V_I)$ | Effective GABA current, inhibitory neurons |
| $J_{AMPA,ext,E}$ (nA) | $g_{E,ext,AMPA}(E_E - V_E)$ | Effective AMPA current, excitatory neurons |
| $J_{AMPA,ext,I}$ (nA) | $g_{I,ext,AMPA}(E_E - V_I)$ | Effective AMPA current, inhibitory neurons |
| $J_{NMDAeff,E}$ (nA) | $\frac{g_{E,rec,NMDA}(E_E - V_E)}{1 + \frac{1}{3.57}e^{-0.062V_E}}$ | Effective NMDA current, excitatory neurons |
| $J_{NMDAeff,I}$ (nA) | $\frac{g_{I,rec,NMDA}(E_E - V_I)}{1 + \frac{1}{3.57}e^{-0.062V_I}}$ | Effective NMDA current, inhibitory neurons |

trials as a separate behavioral outcome different from correct or incorrect decision[32], whereas many other studies assign a choice at random on trials when the network does not reach the decision threshold[2,4,28,44,45]. The random assignment of choices on invalid trials

can conceal differences in network dynamics, making distinct dynamical regimes indistinguishable in psychometric functions[45].

### Phase plane and bifurcation analysis

We analyzed the mean-field model to find null-clines and fixed points using MatLab's fsolve function with the Levenberg-Marquant algorithm and a tolerance of $1 \times 10^{-6}$. To identify the stability of the fixed points, we computed the Jacobian matrix analytically and found its eigenvalues numerically using the eig() function in MatLab. For the saddle points, $\tau_{slow}$ is the inverse of the positive eigenvalue of the Jacobian matrix.

### Recurrent neural network models

Recurrent neural networks (RNNs) were composed of 100 excitatory and 25 inhibitory units. We obtained the same results with networks twice as large (Supplementary Fig. 5). The dynamics of these networks were governed by the equations:

$$\mathbf{x}_E(t) = (1 - \alpha_r)\mathbf{x}_E(t-1) + \alpha_r(\mathbf{W}^{EE}\mathbf{r}_E(t-1) - \mathbf{W}^{IE}\mathbf{r}_I(t-1) + \mathbf{W}^{in}\mathbf{x}_{in}(t) + \boldsymbol{\sigma}_r^E(t)), \tag{26}$$

$$\mathbf{x}_I(t) = (1 - \alpha_r)\mathbf{x}_I(t-1) + \alpha_r(\mathbf{W}^{EI}\mathbf{r}_E(t-1) - \mathbf{W}^{II}\mathbf{r}_I(t-1) + \boldsymbol{\sigma}_r^I(t)), \tag{27}$$

$$\mathbf{x}_{in}(t) = (1 - \alpha_{in})\mathbf{x}_{in}(t-1) + \alpha_{in}\mathbf{u}(t), \tag{28}$$

$$\mathbf{r}_{E(I)}(t) = s_{E(I)}[\mathbf{x}_{E(I)}]_+, \tag{29}$$

$$\mathbf{z}(t) = \mathbf{W}^{out}\mathbf{r}_E(t). \tag{30}$$

Here $\mathbf{x}_E$ and $\mathbf{x}_I$ are the vectors of activation variables for excitatory and inhibitory units, respectively. $\mathbf{r}_E$ and $\mathbf{r}_I$ are the corresponding activities after applying the rectified linear (RELU) nonlinearity $s_{E(I)}[]_+$, where $s_{E(I)}$ sets the excitability of the excitatory or inhibitory units. $\mathbf{x}_{in}$ is the input activation and $\mathbf{u}(t)$ is the instantaneous input. The time constants of recurrent units and inputs are set by $\alpha_r$ and $\alpha_{in}$. Weights within and between units are housed in the matricies $\mathbf{W}_{EE}$, $\mathbf{W}_{EI}$, $\mathbf{W}_{IE}$, $\mathbf{W}_{II}$. Only the excitatory units receive projections from the input and project to the output through $\mathbf{W}^{in}$ and $\mathbf{W}^{out}$, respectively.

RNNs received two input streams $\mathbf{u}(t) = [u_1(t), u_2(t)]$ representing sensory evidence:

$$u_i(t,c) = \begin{cases} u_0 + (1 + \mu\frac{c}{100}) + \sigma_{in,i}(t) & t_{stim,on} < t < t_{stim,off}, \quad i = 1 \\ u_0 + (1 - \mu\frac{c}{100}) + \sigma_{in,i}(t) & t_{stim,on} < t < t_{stim,off}, \quad i = 2 \\ u_0 + \sigma_{in,i}(t) & \text{otherwise}. \end{cases} \tag{31}$$

The stimulus period was 21 time steps and $t_{stim,on}$ and $t_{stim,off}$ were uniquely chosen for each trial. The stimulus magnitude $\mu = 3.2$ was fixed and stimulus difficulty was set by $c$ which ranged between $-20$ and 20.

The recurrent and input noise are modeled by the elements of $\boldsymbol{\sigma}_r^{E(I)}(t)$ and $\boldsymbol{\sigma}_{in}(t)$ that are sampled from a Gaussian distribution. We ensure that each element has a standard deviation $\sigma_{0,r}$ and $\sigma_{0,in}$ via scaling:

$$\sigma_{r,i}^{E(I)}(t) = \sqrt{2\alpha_r}\sigma_{0,r}\mathcal{N}(0,1), \tag{32}$$

$$\sigma_{in,i}(t) = \sqrt{\frac{2}{\alpha_{in}}}\sigma_{0,in}\mathcal{N}(0,1). \tag{33}$$

## RNN training

The goal of RNN training is to minimize the difference between the output $z$ ($N_{trial} \times N_{time} \times N_{out}$) and targets $T$ ($N_{trial} \times N_{time} \times N_{out}$). We set the entries in $T$ to the baseline value of 0.2 and, following a stimulus onset, raise the entries to 1 for the output corresponding to the correct choice. This target is designed to train the network to remain in a low activity state until stimulated and elevate the correct output in response to a stimulus. Half of training trials were catch trials, on which no stimulus was presented and target values remained at 0.2 throughout the trial. The training batch consisted of $N_{trial} = 200$ trials which were randomly generated every training epoch. Within the training batch, noncatch trials were equally divided between possible choices and the difficulty was randomly sampled.

Recurrent network weights were randomly initialized from a Gamma distribution with a shape $w_\mu = 0.0375$ and scale $w_\sigma = 0.5$ for excitatory weights $W^{EE}$, $W^{EI}$, and $\theta w_\mu$ and scale $w_\sigma$ for inhibitory weights $W^{IE}$, $W^{II}$. The scaling factor $\theta = N_E s_E / N_I s_I$ adjusts the strength of inhibitory connections to offset for differences in the number and excitability between excitatory and inhibitory units. Input and output weights $W^{in}$, $W^{out}$ were randomly initialized from a uniform distribution and then values were normalized so the weights associated with each input and output summed to 1 across units. All weights were trained via back-propagation through time to minimize the loss function:

$$\mathcal{L} = \frac{1}{N_{trial}} \frac{1}{N_{time}} \sum_{i=1}^{N_{trial}} \sum_{t=1}^{N_{time}} \left( \frac{1}{N_{out}} \sum_{o=1}^{N_{out}} M_{i,t}(T_{i,t,o} - z_{i,t,o})^2 + \frac{\lambda_x}{N_e + N_i} \sum_{n=1}^{N_e+N_i} x_{i,t,n}^2 \right)$$
$$+ \frac{\lambda_w}{(N_e + N_i)^2} \sum_{m,l=1}^{N_e+N_i} |W_{ml}|.$$

(34)

Here $x$ is a concatenation of $x_E$ and $x_I$ of the size $N_{trial} \times N_{time} \times (N_E + N_I)$, and $W$ is a concatenation of $W^{EE}$, $W^{EI}$, $W^{IE}$, and $W^{II}$ of the size $(N_E + N_I) \times (N_E + N_I)$. To encourage the network to integrate the stimulus for extended time, we used a mask $M$ ($N_{trial} \times N_{time}$), where entries were zero during the stimulus period so that time points during the stimulus were not considered when calculating the error term of the loss function. On catch trials, all entries of $M$ were set to 1. The hyperparameter $\lambda_x = 0.1$ controls the amount of L2 regularization intended to minimize the activation of each unit. The hyperparameter $\lambda_w = 1.0$ controls the amount of L1 regularization applied to weights. We updated the weights by stochastic gradient descent using the ADAM optimizer in PyTorch and Python 3.7 with a learning rate 0.01. During training, the norm of the gradient was clipped at 1.

To maintain the identity of excitatory and inhibitory units and to keep the input and output weights positive, all negative elements of $W^{EE}$, $W^{EI}$, $W^{IE}$, $W^{II}$, $W^{in}$, and $W^{out}$ were set to 0 after every training step. We prevent self-connections by elementwise multiplying $W^{EE}$ and $W^{II}$ by $(1 - I)$, where $I$ is the identity matrix and $1$ is a matrix of 1s, after every training step.

We terminated RNN training based on its task performance. We tested RNN performance on a validation batch of trials after every training epoch. Each validation batch consisted of 100 trials with stimulus strength ranging between −20 and 20 in steps of 2. The network registered a decision when the difference between the output variables was above a threshold of 0.25. Trials were considered valid if at least 75% of the prestimulus period was below the decision threshold and at least 50% of the post stimulus period was above the decision threshold. Overall performance was measured as the fraction of correct choices out of all trials except for the ambiguous case where stimulus was equal to 0. We compute the accuracy and the psychometric function only using valid trials. We terminated training when a network's overall performance reached 85%. RNN parameter values are shown in Table 2.

**Table 2 | Recurrent neural network parameters**

| Parameter | Value | Description |
|---|---|---|
| **RNN parameters** | | |
| $N_E$ | 100 | Number of excitatory units |
| $N_I$ | 25 | Number of inhibitory units |
| $N_{in}$ | 2 | Number of inputs |
| $N_{out}$ | 2 | Number of outputs |
| $N_{time}$ | 60 | Number of time steps in a trial |
| $\alpha_r$ | 0.2 | Recurrent unit time constant |
| $\alpha_{in}$ | 0.2 | Input time constant |
| $s_E$ | $\in [0.5, 1.5]$ | RELU slope, excitatory units |
| $s_I$ | 1 | RELU slope, inhibitory units |
| $u_0$ | 0.2 | Input baseline |
| $\mu$ | 3.2 | Stimulus magnitude |
| $c$ | $\in [-20, 20]$ | Stimulus strength |
| $\sigma_{0,r}$ | 0.35 | Recurrent noise level |
| $\sigma_{0,in}$ | 0.05 | Input noise level |
| **RNN training parameters** | | |
| $N_{trial}$ | 200 | Number of trials in a training epoch |
| $f_{catch}$ | 0.5 | Fraction of training catch trials |
| $\lambda_x$ | 0.1 | Hyperparameter for activation regularization |
| $\lambda_w$ | 1.0 | Hyperparameter for weight regularization |
| $w_\mu$ | 0.0375 | Initial weight distribution shape parameter |
| $w_\sigma$ | 0.5 | Initial weight distribution scale parameter |
| $\theta$ | $\frac{N_E s_E}{N_I s_I}$ | Inhibitory weight scaling factor |

## Measuring choice selectivity of RNN units

After training, we analyzed the activity of excitatory and inhibitory RNN units to quantify their choice selectivity. Our metric is based on the ability to decode the choice registered by the network based on the activity of the unit at the time point immediately following stimulus offset[21]. For each unit, we computed the receiver operating characteristic (ROC) using the roc function and the area under the ROC curve ($AUC_{ROC}$) using the trapz function in Matlab. A unit with the same activity for either choice will have an $AUC_{ROC}$ equal to 0.5, thus our choice selectivity measure was defined by $AUC_{ROC} - 0.5$. To identify significantly selective units, we compared $AUC_{ROC}$ to a shuffled distribution generated from that unit's activity by shuffling the choice outcomes 150 times. We considered units to be choice selective if their $AUC_{ROC}$ fell within the lowest or highest 2.5% percentiles of the shuffled $AUC_{ROC}$ distribution.

## Measuring connection specificity in RNNs

We measured the specificity of connections between choice selective units in RNNs. For each connection class (EE, EI, IE, and II), we computed $\langle w^+ \rangle$ and $\langle w^- \rangle$, the mean strength of the weights between significantly selective units with, respectively, the same and opposite selectivity. Then we computed the specificity $\gamma$ as:

$$\gamma = \frac{\langle w^+ \rangle - \langle w^- \rangle}{\langle w^+ \rangle + \langle w^- \rangle}.$$

(35)

This expression is identical to the specificity $\gamma$ used in the mean-field model. To assess significance of correlations between $\gamma$ for the 4 connection classes, we computed a shuffled distribution constructed by shuffling the network labels 5000 times.

## Perturbing inhibitory populations

We perturbed activity of inhibitory neurons by delivering the same constant input to all inhibitory neurons during the stimulus period. In

**Table 3 | Four-variable mean-field model parameters**

| Parameter | Value | Description |
|---|---|---|
| **RNN parameters** | | |
| $J_{NMDA,E}$ (nA) | 0.4235 | Strength of NMDA synapses onto excitatory neurons |
| $J_{NMDA,I}$ (nA) | 0.5743 | Strength of NMDA synapses onto inhibitory neurons |
| $J_{GABA,E}$ (nA) | −0.4699 | Strength of GABA synapses onto excitatory neurons |
| $J_{GABA,I}$ (nA) | −0.6421 | Strength of GABA synapses onto inhibitory neurons |
| $I_{0,E}$ (nA) | 0.7707 | Background current to excitatory neurons |
| $I_{0,I}$ (nA) | 1.0267 | Background current to inhibitory neurons |

the mean-field model, we modified the parameter $\nu_{0,I}$ by a small amount within the range [−0.5, 0.5] around a baseline. We used two baseline values of $\nu_{0,I}$: 11.5 for low-inhibitory regime and 14 for high-inhibitory regime. In RNNs, we delivered perturbations in a similar manner, where we delivered a constant input within the range [−1, 1] during the stimulus period.

## Four-variable mean-field model

To model the effects of inhibitory-inhibitory specificity and dynamics of inhibitory synapses, we developed a simplified version of our model which explicitly modeled the activity of selective inhibitory populations. In this model, the dynamics of NMDA synapses for excitatory populations $E_1$ and $E_2$ ($i = 1$ and $i = 2$, respectively) are governed by:

$$\frac{dS_i}{dt} = -\frac{S_i}{\tau_{NMDA}} + (1 - S_i)\gamma\Phi(x_i), \quad (36)$$

and dynamics of GABA synapses for inhibitory populations $I_1$ and $I_2$ ($i = 3$ and $i = 4$, respectively) are governed by:

$$\frac{dS_i}{dt} = -\frac{S_i}{\tau_{GABA}} + \Phi(x_i). \quad (37)$$

The nonlinear activation function $\Phi(x)$ is of the form Eq. (2) with $a = 310$ nC$^{-1}$, $b = 125$ Hz, and $c = 0.16$ s for excitatory populations $E_1$ and $E_2$, and $a = 615$ nC$^{-1}$, $b = 177$ Hz, and $c = 0.087$ s for inhibitory populations $I_1$ and $I_2$. The input to population $i$ is

$$x_i = \sum_{j=1}^{4} A_{i,j} S_j + I_{0,E(I)} + I_{stim,i} + I_{\nu,i}, \quad (38)$$

where the adjacency matrix **A** is

$$\mathbf{A} = \begin{pmatrix} w_{EE}^{+} J_{NMDA,E} & w_{EE}^{-} J_{NMDA,E} & w_{IE}^{+} J_{GABA,E} & w_{IE}^{-} J_{GABA,E} \\ w_{EE}^{-} J_{NMDA,E} & w_{EE}^{+} J_{NMDA,E} & w_{IE}^{-} J_{GABA,E} & w_{IE}^{+} J_{GABA,E} \\ w_{EI}^{+} J_{NMDA,I} & w_{EI}^{-} J_{NMDA,I} & w_{II}^{+} J_{GABA,I} & w_{II}^{-} J_{GABA,I} \\ w_{EI}^{-} J_{NMDA,I} & w_{EI}^{+} J_{NMDA,I} & w_{II}^{-} J_{GABA,I} & w_{II}^{+} J_{GABA,I} \end{pmatrix}. \quad (39)$$

Only excitatory populations ($i = 1$ and $i = 2$) receive stimulus information through $I_{stim}$, which is identical to Eq. (4). Noise is introduced by $I_{\nu,i}$ which is implemented as in the two-variable model (Eq. (5)) with the standard deviation of $\nu(t)$ set to 0.2 nA.

The weight parameters $w_{EE}^{+}$, $w_{EE}^{-}$, $w_{EI}^{+}$, $w_{EI}^{-}$, $w_{IE}^{+}$, $w_{IE}^{-}$, $w_{II}^{+}$, and $w_{II}^{-}$ were defined as in the two-variable model. The difference is the addition of $w_{II}^{+}$, and $w_{II}^{-}$ which define the specificity of inhibitory-inhibitory connections and depend on $\gamma_{II}$ which can range between [−1, 1]. The synaptic parameters $J_{NMDA,E}$, $J_{NMDA,I}$, $J_{GABA,E}$, $J_{GABA,I}$, and the background input currents $I_{0,E(I)}$ were chosen so that the firing rate dynamics of $E_1$ and $E_2$ matched that of the two-variable model on a noiseless trial with a stimulus strength of 0.05 using PyABC parameter inference[51]. The values of these parameters are defined in Table 3. Simulations of the four-variable model were performed in Python 3.7.

## Reporting summary

Further information on research design is available in the Nature Portfolio Reporting Summary linked to this article.

## Data availability

The data used in this study can be reproduced using the source code.

## Code availability

The source code to reproduce the results of this study is available on GitHub (https://github.com/engellab/selective-inhibition-models).

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

## Acknowledgements

This work was supported by the NIH grants F32MH123011 (J.P.R.), R01 EB026949 (A.K.C. and T.A.E.), and 2R01EY022979 (A.K.C.), Alfred P. Sloan Foundation Research Fellowship (T.A.E.), and the ISQEB program at the Simons Center for Quantitative Biology at CSHL (J.P.R.). Computer simulations for this work were performed with assistance from the NIH Grant S10OD028632-01. We thank M. Genkin for thoughtful comments on the manuscript.

## Author contributions

J.P.R., A.K.C and T.A.E. designed the research. J.P.R. developed the code and performed computer simulations. J.P.R., A.K.C and T.A.E. wrote the paper.

## Competing interests

The authors declare no competing interests.
