## [Peer Review File · Nature Communications]

Choice selective inhibition drives stability and competition in decision circuitsREVIEWER COMMENTS

Reviewer #1 (Remarks to the Author):

The authors present an in-depth computational study of perceptual decision making, taking into account the recent experimental finding on inhibitory neural selectivity (Najafi et al. 2019) which is not captured by classical neurobiological models of decision making (Wong and Wang 2006). The present study uses both a mean-field analysis and recurrent neural network (RNN) simulations to uncover a dual role of selective inhibitory populations: as elements to drive choice competition and as stabilizers of the network dynamics. The model also indicates that selective inhibition also brings a broader parameter range in which decision making is possible, and the relationship between inhibitory circuits and speed-accuracy tradeoff.

Overall, this study is of high quality and tackles a timing and interesting issue. Especially noteworthy is the use of two complementary theoretical/computational approaches (the mean-field model and the RNN simulations) which are not usually employed together. This study is able to combine both approaches nicely and to deliver interesting results in the process.

I only have a few concerns which would need to be addressed:

1) Although the analysis of the mean-field model is exhaustive, including considerations on E-to-E, E-to-I and I-to-E synapses, I missed a brief analysis of the role of I-to-I synapses. This is discussed later in the text corresponding to the RNN system, but these results lack the clarity which one would achieve with a mean-field analysis. In principle, the presence of I-to-I synapses could translate the competition from excitatory to inhibitory populations --which, after inheriting the choice specificity from the excitatory populations, would engage in a mutually inhibitory, winner-take-all dynamics which will determine the behavioral outcome. Because of this potentially relevant translation of the decision dynamics from excitatory to inhibitory neurons, I think that the authors should extend their mean-field study to the situation in which I-to-I synapses are considered (depending on the interest of the results, this could be either a main or a supplementary figure).

2) Similar results to those reported here also appeared last year in another manuscript (B. Liu, C.-C. Lo and K.-A. Wu, BioRxiv 2021), which also proposed the same circuit (i.e. two excitatory and two inhibitory populations for decision making) to explain the results of Najafi et al. Liu et al. also obtain results linking the inhibitory selectivity structure to speed-accuracy tradeoffs, and other considerations regarding the energy landscape of the system.

I consider that the present work is more complete, especially since it includes the study of RNNs, which Liu et al. do not consider. However, the authors should use this chance to include a detailed comparison between their findings (which are also supported by RNNs) and those of Liu et al. --are their speed-accuracy tradeoff findings in agreement? How does the mean-field differs from that of Liu et al.? A new paragraph or subsection in the discussion could be used for this, where the authors could also capitalize on the differences and improvements of their models over the preliminary ones shown in Najafi et al.

Minor comments:

3) At the beginning of the Results section, I found the introduction of the sigma parameter a bit confusing --as it appear in isolation and then with different indices to denote similar but not identical concepts. I would suggest to revise this notation and maybe use a different symbol for the balance between connection strengths.

4) Line 461: ' τ_{slow} is the positive eigenvalue of the Jacobian matrix'. I wonder whether the authors meant to say 'the inverse of the positive eigenvalue' here?

Reviewer #2 (Remarks to the Author):

The manuscript titled "Two roles for choice selective inhibition in decision-making circuits" investigates the functional contributions of selective inhibitory neurons underlying perceptual decision-making by using computational modelling of attractor networks and theoretical analysis to show that selective inhibition can maximise competition and stabilise strongly connected excitatory neurons during decision-making. The study was inspired by recent experimental findings. The manuscript was generally well written.

Context wise, it should be noted that previous studies have already shown (including stability analysis) the role of inhibition in balancing excitation (e.g. Eckhoff, Wong-Lin and Holmes, *J. Neurosci.*, 2009; Niyogi and Wong-Lin, *PLoS Comput. Biol.*, 2013; Lam, Borduqui et al., *J. Neurosci.*, 2022) to enable robust decision-making. Other previous work had also investigated selective inhibitory neurons by explicitly modelling them (e.g. Najafi, Elsayed et al., *Neuron*, 2020; Mahajan and Mysore, *Nat. Commun.*, 2022) or implicitly modelling them (e.g. Mazurek et al., *Cereb. Cortex*, 2003). In my opinion, the main contribution in this work was the theoretical analysis using more biologically realistic nonlinear (mean-field) model with explicit selective inhibitory neurons.

Major queries and comments:

1. The work had two explicit selective excitatory neural populations modelled, while implicitly modelled 3 (2 selective and 1 non-selective) inhibitory neural populations and 1 non-selective neural population. If there was already 2 selective inhibitory neural populations involved, it was unclear why the non-selective inhibitory neural population was needed (e.g. Najafi, Elsayed et al., *Neuron*, 2020; Mahajan and Mysore, *Nat. Commun.*, 2022) (other than due to legacy from past modelling work - e.g. Wong and Wang, *J. Neurosci.*, 2006). Moreover, the interactions between the two selective excitatory neural population should be excitatory (e.g. Najafi, Elsayed et al., *Neuron*, 2020) rather than inhibitory. Where were the connections between the inhibitory neural populations, and self-inhibition? This model architectural issue constitutes a major problem with this work. To provide better clarity, the authors should perhaps consider redoing the work at least without this non-selective inhibitory neural population, and consider excitatory connections between the two selective excitatory neural populations.

2. Given the importance of the selective inhibitory neurons, it was unclear why the authors did not model the dynamics of the (averaged) GABA-mediated synapses (e.g. Najafi, Elsayed et al., *Neuron*, 2020) within the mean-field framework (e.g. Wong and Wang, *J. Neurosci.*, 2006). This will allow all the selective neural populations to be on the same footing and with fairer comparison/analysis.

3. Another limitation of this study was that it did not compare with experimental data. There were too many invalid trials in both the mean-field model and the RNN model. I do not think such high numbers could be observed in experiments (with well-trained participants). The authors considered the invalid trials as incorrect trials - this is not the right way to do it, but invalid trials should be discarded.

4. The effect of contra- and ipsi-specific selective inhibitions on the stability diagram was relatively trivial - essentially depending on where the system was in the stability diagram. Hence, this may not be too novel theoretically.

5. The perturbation in the RNN part of the work was inconsistent and indirect. Excitatory neurons were perturbed by increasing their (input-output) gains while separately, the inhibitory neurons were perturbed via some constant inputs - inconsistent. Hence, it was unclear why the types of perturbations were not the same for excitatory and inhibitory neurons. Why did the authors not set the excitatory/inhibitory neurons' average connections at a higher/lower strength, prior to training, to see the consequences of the trained RNN.

6. Why were there only 125 neurons used in the RNN and not higher (mean-field has higher number)? Why were self-connections not considered in the RNN, but were considered in the mean-field model?

7. The trained RNN showed a higher fraction of neurons with significant choice selectivity for inhibitory than excitatory neurons, and that inhibitory neurons had overall higher choice selectivity than excitatory neurons. Was this observed in experiment(s)? Didn't the experiment by e.g. Najafi, Elsayed et al. (2020) show that they have about the same choice selectivity?

Minor/specific comments and suggestions:

1. Title. Too general; perhaps specify the roles.

2. Abstract. 'altering'. Specify how.

3. Line 26. 'highly'. I am not so sure about this.

4. Lines 46-47, and elsewhere in the manuscript. 'inhibition drives competition'. This is not new. Perhaps there is a better way to state the novelty (see above)?

5. Line 72, and elsewhere in the manuscript. Why use sigma (which is usually reserved for mathematical summation), and superscript in the sigmas (which is usually reserved for power in mathematics)?

6. Line 96. 'eight fixed points'. This is confusing. In this work, the maximum number of fixed points for one phase plane was just five.

7. Line 157. Should be 'contraspecific'. Spelling error.

8. Lines 171-174. This is well known. Perhaps there is a better way to state the novelty?

9. Lines 176-178. Persistence does not really indicate 'good' decision-making. Decision can still be made without (working) memory. See the slow integration (and perhaps accurate decision) in e.g. Wong and Wang (2006).

10. Lines 200-202. This is not right. Invalid trials should not be considered as incorrect. (See above.) The standard deviation seemed too large, almost the same as the mean.

11. Line 204. Should be 'trials'. Spelling error.

12. Line 212-215. Should compare with experimental data. (See above.)

13. Line 262. Should be 'mean-field'. Spelling error.

14. Generally throughout the manuscript. The term 'reaction time' should probably be replaced by 'decision time' in text, figures and tables, as non-decision latency was not included here. If experimental data was to be compared (which the authors should), reaction time should be used with non-decision latency included.

15. Line 303. '... broaden ...'. This is not really much of a broadening, but just finely balancing/compensatory mechanism, moving across different points on the same stability diagram. E.g. the green strips in Fig. 2c remain about the same in width. A true broadening of the decision regime can be seen from e.g. Niyogi and Wong-Lin (2013).

16. Fig. 3. The decision threshold was unclear. In Fig. 3d, the decision thresholds were not fixed in values, but for previous modelling studies they are fixed numbers (e.g. 15 Hz in Wong and Wang, 2006). In experiments, the peaks of the neuronal ramping activities were also relatively constant (e.g. see Roitman and Shadlen, *J. Neurosci.*, 2002). In Fig. 3g, the stability was just the network reaching closer to a bifurcation point (and slowing the ramping activity); theoretically, this was not new.

17. Fig. 4. Should show the choice accuracy and decision time here.

18. Fig. 6c. This could possibly be just the sweeping across two bifurcation points within the same decision-making regime (e.g. see Wong and Wang, 2006). If so, mention/explain in main text.

19. Supplementary Fig. 1. The figure title should be edited as some of the phase planes (panels b and d) do allow decision making. To some extent, panel c was also making ('impulsive') decision (e.g. see Eckhoff, Wong-Lin and Holmes, 2009) but without integrating the evidence well.

20. Supplementary Fig. 6. Not sure whether this is correct, as there was non-selective inhibition between the two selective excitatory neural populations.

21. Methods. For open and transparent science, codes should be made available at least for reviewing, if the journal permits.

Point-by-point responses.

We thank the reviewers for providing thoughtful comments. We addressed all issues that were previously raised in the revised manuscript. We considered the selectivity of inhibitory-to-inhibitory connections, evaluated a four-variable version of the model that accounts for dynamics of inhibitory synapses, and revised the model description to improve clarity. Finally, we included RNNs that were twice the size of our original RNNs and found a similar performance and circuit structure as we originally reported. These revisions, which we believe have strengthened our results, are described in detail in the point-by-point responses. The line numbers refer to the lines in the revised manuscript without track changes.

Reviewer #1:

The authors present an in-depth computational study of perceptual decision making, taking into account the recent experimental finding on inhibitory neural selectivity (Najafi et al. 2019) which is not captured by classical neurobiological models of decision making (Wong and Wang 2006). The present study uses both a mean-field analysis and recurrent neural network (RNN) simulations to uncover a dual role of selective inhibitory populations: as elements to drive choice competition and as stabilizers of the network dynamics. The model also indicates that selective inhibition also brings a broader parameter range in which decision making is possible, and the relationship between inhibitory circuits and speed-accuracy tradeoff.

Overall, this study is of high quality and tackles a timing and interesting issue. Especially noteworthy is the use of two complementary theoretical/computational approaches (the mean-field model and the RNN simulations) which are not usually employed together. This study is able to combine both approaches nicely and to deliver interesting results in the process. I only have a few concerns which would need to be addressed.

Reply:

We thank the reviewer for the positive assessment and suggestions that helped us improve the paper.

1) Although the analysis of the mean-field model is exhaustive, including considerations on E-to-E, E-to-I and I-to-E synapses, I missed a brief analysis of the role of I-to-I synapses. This is discussed later in the text corresponding to the RNN system, but these results lack the clarity which one would achieve with a mean-field analysis. In principle, the presence of I-to-I synapses could translate the competition from excitatory to inhibitory populations --which, after inheriting the choice specificity from the excitatory populations, would engage in a mutually inhibitory, winner-take-all dynamics which will determine the behavioral outcome. Because of this potentially relevant translation of the decision dynamics from excitatory to inhibitory neurons, I think that the authors should extend their mean-field study to the situation in which I-to-I synapses are considered (depending on the interest of the results, this could be either a main or a supplementary figure).

Reply:

We agree with the reviewer that the analysis of the mean-field model would be improved by the consideration of specific inhibitory-to-inhibitory (I-I) connections. In the revised manuscript, we include a four-variable mean-field model which explicitly models the activity of two choice-selective inhibitory populations and two choice-selective excitatory populations. We used this model to examine the effect of specific I-I connections (new Supplementary Fig. 3). We find that circuits with competitive I-I specificity are generally hyper-competitive and reach a decision before the stimulus onset. Overall, inhibitory-to-inhibitory connection specificity had a weaker impact on the presence of the fixed points required for decision making than inhibitory-to-excitatory specificity. We describe these results on lines 154-168 in the revised manuscript.

2) Similar results to those reported here also appeared last year in another manuscript (B. Liu, C.-C. Lo and K.-A. Wu, BioRxiv 2021), which also proposed the same circuit (i.e. two excitatory and two inhibitory populations for decision making) to explain the results of Najafi et al. Liu et al. also obtain results linking the inhibitory selectivity structure to speed-accuracy tradeoffs, and other considerations regarding the energy landscape of the system.

I consider that the present work is more complete, especially since it includes the study of RNNs, which Liu et al. do not consider. However, the authors should use this chance to include a detailed comparison between their findings (which are also supported by RNNs) and those of Liu et al. --are their speed-accuracy tradeoff findings in agreement? How does the mean-field differs from that of Liu et al.? A new paragraph or subsection in the discussion could be used for this, where the authors could also capitalize on the differences and improvements of their models over the preliminary ones shown in Najafi et al.

Reply:

We agree that Liu et al. 2021 study is highly relevant to our work and deserves a comparison with our results. This work focused on isolating the contribution of GABA dynamics to decision-making in circuits with selective inhibition using a mean-field framework similar to ours. The key difference between our approach and that of Liu et al. is that we explore a wide range of circuit configurations defined by inhibitory connectivity motifs under a steady-state assumption for GABA dynamics, whereas Liu et al. consider a narrow space of circuits but model the GABA dynamics explicitly. The steady-state assumption for GABA dynamics is based on the timescale separation between decay time constants of slow NMDA (~100ms) and fast GABA (~5ms) conductance. The slow NMDA dynamics dominate the time evolution of network activity, and one can assume that all other variables reach their steady-state, leading to a reduced two-variable mean-field model (Wong & Wang, 2006, see our response to point 2 of Reviewer #2). Liu et al. considered a set of circuits with parameters chosen so that when the steady-state assumption is applied, all models reduce to the two-variable model with the exact same parameter set. Thus, all differences in dynamics of these circuits were driven by GABA dynamics. In these circuits, the GABA dynamics mediated a speed-accuracy trade-off and, moreover, this tradeoff was more efficient in circuits with selective inhibition. Liu et al. considered a narrow space of circuits that all map onto a single parameter set of a two-variable model and limited their analyses only to ipsi-specific inhibitory connectivity. In contrast, our work explores a wide range of circuit configurations ranging from contraspecific to ipsi-specific inhibitory motifs. Thus, our results and those of Liu et al. are complementary, showing that inhibitory connectivity motifs and GABA dynamics both affect decision-making behavior. We added a comparison between our results and those of Liu et al. in the revised Discussion on lines 437-453.

In the revised manuscript, we also developed a mean-field model with four dynamical variables: 2 NMDA variables for selective excitatory populations and 2 GABA variables for selective inhibitory populations. In this model, we systematically explored the space of circuit configurations ranging from contraspecific to ipsi-specific inhibitory motifs and found results consistent with the two-variable model (new Supplementary Fig. 3). Thus, our findings are robust to the steady-state approximation of GABA dynamics in the two-variable model.

Minor comments:

3) *At the beginning of the Results section, I found the introduction of the sigma parameter a bit confusing --as it appear in isolation and then with different indices to denote similar but not identical concepts. I would suggest to revise this notation and maybe use a different symbol for the balance between connection strengths.*

Reply:

We changed the notation and revised the section introducing the connection specificity parameters to improve clarity on lines 69-90 in the revised manuscript.

4) *Line 461: 'tau_slow is the positive eigenvalue of the Jacobian matrix'. I wonder whether the authors meant to say 'the inverse of the positive eigenvalue' here?*

Reply:

Yes, and we have corrected this mistake.

Reviewer #2:

The manuscript titled "Two roles for choice selective inhibition in decision-making circuits" investigates the functional contributions of selective inhibitory neurons underlying perceptual decision-making by using computational modelling of attractor networks and theoretical analysis to show that selective inhibition can

maximise competition and stabilise strongly connected excitatory neurons during decision-making. The study was inspired by recent experimental findings. The manuscript was generally well written.

Context wise, it should be noted that previous studies have already shown (including stability analysis) the role of inhibition in balancing excitation (e.g. Eckhoff, Wong-Lin and Holmes, J. Neurosci., 2009; Niyogi and Wong-Lin, PLoS Comput. Biol., 2013; Lam, Borduqui et al., J. Neurosci., 2022) to enable robust decision-making. Other previous work had also investigated selective inhibitory neurons by explicitly modelling them (e.g. Najafi, Elsayed et al., Neuron, 2020; Mahajan and Mysore, Nat. Commun., 2022) or implicitly modelling them (e.g. Mazurek et al., Cereb. Cortex, 2003). In my opinion, the main contribution in this work was the theoretical analysis using more biologically realistic nonlinear (mean-field) model with explicit selective inhibitory neurons.

Reply:

We thank the reviewer for careful reading of our manuscript and detailed feedback that helped us improve the paper. We agree with the reviewer that the major contribution of our work is to identify the impact of selective inhibitory neurons on decision-making behavior in nonlinear recurrent network models. The attractor-based mean-field model allows us to understand the effects of selective inhibition on behavior using well established dynamical systems principles. Analysis of RNNs, in addition to the mean-field model, strengthens our results by showing that selective inhibition in heterogeneous networks has the same competitive and stabilizing roles we identify in the mean-field models and these networks leverage selective inhibition to learn the task via backpropagation through time.

Major queries and comments:

1) The work had two explicit selective excitatory neural populations modelled, while implicitly modelled 3 (2 selective and 1 non-selective) inhibitory neural populations and 1 non-selective neural population. If there was already 2 selective inhibitory neural populations involved, it was unclear why the non-selective inhibitory neural population was needed (e.g. Najafi, Elsayed et al., Neuron, 2020; Mahajan and Mysore, Nat. Commun., 2022) (other than due to legacy from past modelling work - e.g. Wong and Wang, J. Neurosci., 2006). Moreover, the interactions between the two selective excitatory neural population should be excitatory (e.g. Najafi, Elsayed et al., Neuron, 2020) rather than inhibitory. Where were the connections between the inhibitory neural populations, and self-inhibition? This model architectural issue constitutes a major problem with this work. To provide better clarity, the authors should perhaps consider redoing the work at least without this non-selective inhibitory neural population, and consider excitatory connections between the two selective excitatory neural populations.

Reply:

Our mean-field model accounts for interactions among 6 populations: 3 excitatory (2 choice selective and 1 nonselective) and 3 inhibitory (2 choice selective and 1 nonselective). Including nonselective neurons in the model is indeed consistent with the previous work (e.g., Wong & Wang 2006) and, moreover, reflects the experimental observation that only a fraction of all recorded neurons shows selectivity for any particular task. This feature of cortical responses is common across many areas and tasks. In particular, Najafi et al. 2020 found that only 13% of excitatory and 16% of inhibitory neurons showed choice selectivity. From the modeling perspective, the fraction of selective neurons is an important parameter controlling circuit dynamics. Varying this parameter while holding other parameters fixed can change the configuration of fixed points and hence the circuit's ability to support decision making. In the revised manuscript, we study the effect of the relative size of selective populations on the fixed points (new Supplementary Fig. 2). This new analysis shows that contraspecific circuits support decision-making dynamics for a wider range of the selective population size, primarily due to a wider parameter range where the working memory attractors are present in the unstimulated circuit.

Our model accounts for all connections among all 6 populations and respects Dayle's law. In particular, the connections between excitatory populations E1 and E2 are excitatory, and the model includes inhibitory-to-inhibitory connections. We clarified the model architecture in the revised manuscript (see our response to reviewer 1 point 3).

2) *Given the importance of the selective inhibitory neurons, it was unclear why the authors did not model the dynamics of the (averaged) GABA-mediated synapses (e.g. Najafi, Elsayed et al., Neuron, 2020) within the mean-field framework (e.g. Wong and Wang, J. Neurosci., 2006). This will allow all the selective neural populations to be on the same footing and with fairer comparison/analysis.*

Reply:

Our mean-field framework reduces the dynamics of the full network with 6 excitatory and inhibitory populations to a two-variable system, following the approach of Wong & Wang 2006. The advantage of the two-dimensional model is that it enables theoretical analysis of fixed points in a two-dimensional phase plane. The model reduction to two dimensions uses several approximations, in particular, it leverages the timescale separation between decay time constants of the slow NMDA (~100ms) and fast GABA (~5ms) and AMPA (~2ms) conductance. The slow NMDA dynamics dominate the time evolution of the system, and one can assume that all other variables reach their steady-state nearly instantly. Note that although we use the steady-state approximation for GABA dynamics, all excitatory and inhibitory neural populations are on the same footing because all populations receive inhibitory inputs via GABA synapses.

The reviewer is right that despite being fast, the dynamics of GABA synapses can affect decision-making behavior (see also our response to point 2 of reviewer 1). Prompted by the reviewer's comment, we investigated the effect of GABA dynamics on decision-making in circuits with selective inhibition. We developed a mean-field model with four dynamical variables: 2 NMDA variables for selective excitatory populations and 2 GABA variables for selective inhibitory populations. In this model, we systematically explored the space of circuit configurations ranging from contraspecific to ipsispecific inhibitory motifs and found results consistent with the two-variable model (new Supplementary Fig. 3). Specifically, the four-variable model also showed the speed-accuracy trade-off when varying the inhibitory specificity and a linear relationship between excitatory and inhibitory specificity in circuits that support decision-making. Thus, our findings of how inhibitory circuit motifs affect decision-making dynamics are robust to the steady-state approximation of GABA dynamics in the two-variable model.

3) *Another limitation of this study was that it did not compare with experimental data. There were too many invalid trials in both the mean-field model and the RNN model. I do not think such high numbers could be observed in experiments (with well-trained participants). The authors considered the invalid trials as incorrect trials - this is not the right way to do it, but invalid trials should be discarded.*

Reply:

Our models (both the mean-field and RNNs) fail to reach the imposed decision threshold on a fraction of trials with low stimulus strength, which we call invalid trials. This behavior, however, is not specific to our models but is common across spiking (Wang 2002; Lam et al. 2021), mean-field (Wong & Wang 2006; Liu et al. 2021) and RNN (Song et al. 2016) models of decision-making. Our treatment of invalid trials is more conservative than in many other studies, as we report invalid trials as a separate behavioral outcome different from correct or incorrect decision (Eckhoff et al. 2009), whereas most other studies assign a choice at random on trials when the network does not reach the decision threshold (Wang 2002; Wong & Wang 2006; Song et al. 2016; Lam et al. 2021; Liu et al. 2021). The random assignment of choices on invalid trials can conceal differences in network dynamics, making distinct dynamical regimes indistinguishable in psychometric functions (Lam et al. 2021). For example, networks with elevated versus lowered E/I balance have different speeds of dynamics but identical psychometric functions if choices are assigned at random on invalid trials (Lam et al. 2021). These dynamical regimes are distinguishable if separating the invalid trials, since networks with lowered E/I balance produce more invalid trials but are more accurate when reaching the decision threshold than networks with elevated E/I balance. We have added a more complete description of our handling of trials in the Methods section (lines 533-543).

We agree with the reviewer that circuit models frequently differ from experimental subjects in the rate of trial completion, which has been attributed to an urgency signal gating the evidence accumulation process which is absent in most circuit models (Churchland et al., 2008; Cisek et al., 2009; Thura et al., 2012; Carland et al., 2015). One possible mechanism for an urgency signal in decision circuits could be a nonspecific external ramping input

(Finkelstein et al. 2021). Incorporating such inputs into future models of decision-making would be an important next step in the study of selective inhibition. We explained the reasons for considering the rate of trial completion in the Discussion on lines 454-467.

4) The effect of contra- and ipsi-specific selective inhibitions on the stability diagram was relatively trivial - essentially depending on where the system was in the stability diagram. Hence, this may not be too novel theoretically.

Reply:

Our work connects inhibitory circuitry to a well established dynamical principle in decision-making circuits. It is indeed known that dynamics around the saddle point sets the decision time and accuracy in the attractor circuit model, but the link between this phenomenon and selective inhibition is novel. A major strength of our work is to draw on this well established result to interpret the role of selective inhibition. We clarified the novelty of our work and its relationship to the previous within the revised Discussion (lines 415-427).

5) The perturbation in the RNN part of the work was inconsistent and indirect. Excitatory neurons were perturbed by increasing their (input-output) gains while separately, the inhibitory neurons were perturbed via some constant inputs - inconsistent. Hence, it was unclear why the types of perturbations were not the same for excitatory and inhibitory neurons. Why did the authors not set the excitatory/inhibitory neurons' average connections at a higher/lower strength, prior to training, to see the consequences of the trained RNN.

Reply:

The reviewer refers to two different sets of simulation experiments we performed with RNNs, which address two separate questions and are described in different sub-sections of Results. First, we tested whether the inhibitory circuitry in RNNs can adjust through the training process to compensate for higher or lower excitability of excitatory units. To this end, we set the activation function slope of excitatory units in RNNs to a high (low) value throughout the training process and quantified the circuit structure that emerged after training. We found that RNNs training compensated for higher/lower gain of excitatory units mainly by adjusting the excitatory connection specificity without changing the distribution of inhibitory specificity. This result is consistent with the mean-field model, in which the excitatory specificity has a larger effect than inhibitory specificity to compensate for changes in neural parameters.

Second, we used perturbations of inhibitory unit activity to identify the dynamical regime in which trained RNNs operate. The mean-field model predicted the existence of two dynamical regimes, in which either the stabilizing or competitive role of inhibition dominates. These regimes can be dissociated by measuring changes in behavioral metrics under perturbations of inhibitory activity. We found that our RNNs operated in the stabilizing inhibition regime. Our perturbations of inhibitory unit activity in RNNs replicate an experimentally plausible perturbation on the timescale of a single trial, e.g., with optogenetics. These results show a possibility of determining whether the circuit operates in a stabilizing or competitive inhibition regime via experimentally plausible perturbations of inhibitory neuron activity.

We set the initial conditions for the RNN weights so that inhibitory weights were four times larger than excitatory weights to account for the inhibitory population being 25% the size of the excitatory population, following Song et al. 2016. This approach initializes each RNN to a balanced state with stable recurrent dynamics.

6) Why were there only 125 neurons used in the RNN and not higher (mean-field has higher number)? Why was self-connections not considered in the RNN, but was considered in the mean-field model?

Reply:

We chose 125 units in the RNNs following previous work by Song et al. 2016 on which we based our RNN models. This size proved sufficiently large to perform the task and train a large number of networks in a reasonable time. In the revised manuscript, we confirm that our results are not specific to this RNN size. We trained an additional set of RNNs at twice the size (250 units) and found that our results hold in these larger networks (new Supplementary Fig. 5).

The difference in self-connections between the mean-field and RNN models is due to the fact that dynamical variables in these models are not equivalent. RNN units represent activity of individual neurons, whereas the dynamical variables in the mean-field model represent the average activity of populations of neurons. The self-connections in the mean-field model represent the average feedback input a neuron receives from other neurons in the same population, these are not connections of a neuron onto itself.

7) *The trained RNN showed higher fraction of neurons with significant choice selectivity for inhibitory than excitatory neurons, and that inhibitory neurons had overall higher choice selectivity than excitatory neurons. Was this observed in experiment(s)? Didn't the experiment by e.g. Najafi, Elsayed et al. (2020) showed that they have about the same choice selectivity?*

Reply:

We chose to use RNNs precisely because of their differences with the mean-field model and in turn the brain. The utility of the RNNs is that they are trained through backpropagation through time to perform the task and they allowed us to demonstrate the function of selective inhibition in a heterogeneous network where the circuitry was not designed by hand. The learning process produces a specific circuit motif out of simplified units and while RNNs are approximations of what may happen in the brain we cannot expect them to be an exact match to brain circuitry or functional measurements of activity. It is not surprising that RNNs trained to perform a single task on a simplified stimulus end up with most units being selective for that task. We clarify this difference between RNNs and experimental data in the revised Discussion (lines 399-407).

Minor/specific comments and suggestions:

1) *Title. Too general; perhaps specify the roles.*

Reply:

We have edited the title to specify the roles. It now reads “*Choice selective inhibition drives stability and competition in decision circuits*”.

2) *Abstract. 'altering'. Specify how.*

Reply:

We modified the text to read “*The specificity of inhibitory outputs sets the trade-off between speed and accuracy of decisions by either stabilizing or destabilizing the saddle point dynamics underlying decisions in the circuit.*”

3) *Line 26. 'highly'. I am not so sure about this.*

Reply:

We removed the word to reduce any confusion.

4) *Lines 46-47, and elsewhere in the manuscript. 'inhibition drives competition'. This is not new. Perhaps there is a better way to state the novelty (see above)?*

Reply:

While driving the competition and stability are established roles for inhibition in neural circuits, our work identifies how these roles must be balanced in functional circuits and how selective inhibition can help strike that balance. We revised the text to better reflect the novelty of our work (lines 45-49).

5) *Line 72, and elsewhere in the manuscript. Why use sigma (which is usually reserved for mathematical summation), and superscript in the sigmas (which is usually reserved for power in mathematics)?*

Reply:

We changed our notation of the selectivity parameter throughout the text to avoid confusion.

6) Line 96. 'eight fixed points'. This is confusing. In this work, the maximum number of fixed points for one phase plane was just five.

Reply:

Eight fixed points refers to the total number of fixed points across the stimulated and unstimulated phase planes. We revised the text to clarify this issue (line 104-106): “These dynamics are governed by eight fixed points across the unstimulated and stimulated phase plane which are essential for the functional decision-making and working memory behavior.”

7) Line 157. Should be 'contraspecific'. Spelling error.

Reply:

We have fixed the typo.

8) Lines 171-174. This is well known. Perhaps there is a better way to state the novelty?

Reply:

The fact that the saddle point can bifurcate into a stable fixed point is indeed a known feature of attractor models of decision making. Our novel contribution is to show how the inhibitory circuit structure can control this bifurcation. The text in question “This bifurcation leads to the system stabilizing in a state where firing rates of two choice-selective populations do not sufficiently separate on neutral and difficult stimuli trials, a state where the circuit fails to produce a decision” provides the important context about what this bifurcation means for the circuit behavior. The novelty of our result is in showing that this bifurcation can occur when varying the inhibitory circuit motif. We clarified this issue in the revised Discussion (lines 415-427).

9) Lines 176-178. Persistence does not really indicate 'good' decision-making. Decision can still be made without (working) memory. See the slow integration (and perhaps accurate decision) in e.g. Wong and Wang (2006).

Reply:

We chose to include a working memory component in our criteria as to what circuits support decision-making because it is a feature of many circuit models (e.g., Wang, 2002; Wong & Wang, 2006). We revised the text to explain the motivation for including a working memory requirement for a decision-making circuit (lines 102-104): “Persistence of the decision after stimulus offset allows for a choice readout to be made even after a significant delay and its utility led us to include the working memory of a choice in our criteria for inclusion as a circuit supporting decision-making”.

10) Lines 200-202. This is not right. Invalid trials should not be considered as incorrect. (See above.) The standard deviation seemed too large, almost the same as the mean.

Reply:

We do not consider invalid trials as incorrect, the confusion was due to unclear description in the original manuscript. We clarified this issue in the revised text (lines 231-232): “We trained networks until the correct choice was made on 85% of all trials (including correct, error, and invalid trials) in a 200 trial epoch.” The mean and standard deviation reported in the initial manuscript contained a typo, the mean was reported as 10,4343 with a misplaced comma, it should read 104,343. We corrected the typo.

11) Line 204. Should be 'trials'. Spelling error.

Reply:

We fixed the typo.

12) Line2 212-215. Should compare with experimental data. (See above.)

Reply:

We addressed this point in the Discussion and revised the text to compare with experimental data (lines 250-251): “In this respect RNNs differ from experimental data in which excitatory and inhibitory neurons contained similar choice information²¹”. (also see our response to major point 7).

13) Line 262. Should be 'mean-field'. Spelling error.

Reply:

We fixed this typo.

14) Generally throughout the manuscript. The term 'reaction time' should probably be replaced by 'decision time' in text, figures and tables, as non-decision latency was not included here. If experimental data was to be compared (which the authors should), reaction time should be used with non-decision latency included.

Reply:

We changed reaction time to “decision time” throughout the manuscript to reflect that we do not add a non-decision latency in our model.

15) Line 303. '... broaden ...'. This is not really much of a broadening, but just finely balancing/compensatory mechanism, moving across different points on the same stability diagram. E.g. the green strips in Fig. 2c remain about the same in width. A true broadening of the decision regime can be seen from e.g. Niyogi and Wong-Lin (2013).

Reply:

Yes, the range of the EE specificity parameter is the same for any given level of inhibitory specificity. We revised the text to clarify this issue (lines 337-339): “In the mean-field model, choice selective inhibition and specific connections from inhibitory to excitatory populations expand the excitatory-excitatory specificity parameter space of circuits that support decision-making.”

16) Fig. 3. The decision threshold was unclear. In Fig. 3d, the decision thresholds were not fixed in values, but for previous modelling studies they are fixed numbers (e.g. 15 Hz in Wong and Wang, 2006). In experiments, the peaks of the neuronal ramping activities were also relatively constant (e.g. see Roitman and Shadlen, J. Neurosci., 2002). In Fig. 3g, the stability was just the network reaching closer to a bifurcation point (and slowing the ramping activity); theoretically, this was not new.

Reply:

We set the decision-threshold to be a 15Hz difference between the firing rates of excitatory populations to account for differences in the baseline firing rate of unstimulated circuits, i.e. we account for the possibility that the low-activity symmetrical attractor shifts across circuit configurations. Fig. 3d makes it clear that this modeling choice has a minimal effect on the location of the decision threshold relative to the choice attractors. In Fig. 3g, we show that as the connectivity motif shifts from contra- to ipsi-specific, the circuit moves closer to the bifurcation point. The novelty of this result is in establishing that the inhibitory circuit motif controls this known bifurcation mechanism. We clarified these issues in the Results (lines 189 to 205) and Discussion (lines 415 to 427) in the revised manuscript.

17) Fig. 4. Should show the choice accuracy and decision time here.

Reply:

We show choice accuracy and decision time in Fig. 3 to frame the saddle point analysis showing that the inhibitory circuit structure controls the bifurcation of the symmetrical high-activity fixed point in the stimulated phase plane. Fig. 4, on the other hand, shows that the inhibitory circuit motif controls the working memory attractors in the unstimulated phase plane, which is complementary to Fig. 3.

18) Fig. 6c. This could possibly be just the sweeping across two bifurcation points within the same decision-making regime (e.g. see Wong and Wang, 2006). If so, mention/explain in main text.

Reply:

The reviewer is right that the U-shape dependence of τ_{slow} on the baseline input to inhibitory neurons results from the system approaching bifurcation points at either extreme of the parameter range that supports decision making. This U-shaped dependence gives rise to two dynamical regimes in which either stabilizing or competitive role of inhibition dominates. We clarified this point in the revised manuscript (line 310-313): “This U-shape dependence of τ_{slow} on the baseline input to inhibitory neurons $v_{0,I}$ results from the system approaching bifurcation points at either extreme of the parameter range that supports decision making⁴ (Supplementary Fig. 10)” and added bifurcation diagrams with the baseline input to inhibitory neurons as a control parameter (new Supplementary Fig. 10).

19) *Supplementary Fig. 1. The figure title should be edited as some of the phase planes (panels b and d) do allow decision making. To some extent, panel c was also making ('impulsive') decision (e.g. see Eckhoff, Wong-Lin and Holmes, 2009) but without integrating the evidence well.*

Reply:

We revised the figure title: “Examples of circuits lacking fixed points necessary for decision making or working memory”. Models lacking a stable low-activity fixed point (as in panel c) cannot be considered as making impulsive decisions without integrating evidence well. These models settle in one of the asymmetric attractors in the absence of stimulus and noise, and due to the lack of an undecided state we do not interpret their dynamics as a decision process. In contrast, models in which the low-activity attractor is stable but close to the bifurcation reside in the symmetric state and can generate “impulsive” decisions driven by noise on a fraction of trials (Eckhoff et al. 2009).

20) *Supplementary Fig. 6. Not sure whether this is correct, as there was non-selective inhibition between the two selective excitatory neural populations.*

Reply:

This figure shows that changing the specificity of excitatory-excitatory connections (defined by γ_{EE} , y-axis) has a larger effect on performance and saddle point dynamics than the specificity of inhibitory output connections (defined by γ_{IE} , x-axis). The selectivity of inhibitory firing rates is defined by γ_{EI} and is set to 0.25 in these circuits. At each point on these plots the strength of connections between nonselective populations and selective populations are unchanged. We have updated the discussion of this figure (lines 289-290) to read “For accuracy, decision-time and τ_{slow} , changes in γ_{EE} are far more effective than changes in inhibitory specificity (Supplementary Fig. 9) when all other parameters are held constant.”

21) *Methods. For open and transparent science, codes should be made available at least for reviewing, if the journal permits.*

Reply:

We wholeheartedly support open and transparent science and release all our code on GitHub with every publication. We will release a GitHub repository with the models described in this work upon publication of the manuscript. We included all code as an attachment to this revision for your review. We will clean up the code and write a detailed documentation prior to public release.

REVIEWERS' COMMENTS

Reviewer #1 (Remarks to the Author):

In this revised version of the manuscript, the authors have satisfactorily addressed my previous concerns, including the role of I-to-I interactions and relationship with other studies. Together with the improvements regarding the suggestions by the other reviewer, I think that this work has significantly improved in quality and I can now recommend its publication.

Jorge Mejias (voluntary signature)

Reviewer #2 (Remarks to the Author):

The authors have responded satisfactorily to my queries and comments.

They have done a commendable job in revising their manuscript, and overall improving the paper.